# Sparse Spiking Gradient Descent

**Nicolas Perez-Nieves**
Electrical and Electronic Engineering
Imperial College London
London, United Kingdom
nicolas.perez14@imperial.ac.uk

**Dan F.M. Goodman**
Electrical and Electronic Engineering
Imperial College London
London, United Kingdom
d.goodman@imperial.ac.uk

## Abstract

There is an increasing interest in emulating Spiking Neural Networks (SNNs) on neuromorphic computing devices due to their low energy consumption. Recent advances have allowed training SNNs to a point where they start to compete with traditional Artificial Neural Networks (ANNs) in terms of accuracy, while at the same time being energy efficient when run on neuromorphic hardware. However, the process of training SNNs is still based on dense tensor operations originally developed for ANNs which do not leverage the spatiotemporally sparse nature of SNNs. We present here the first sparse SNN backpropagation algorithm which achieves the same or better accuracy as current state of the art methods while being significantly faster and more memory efficient. We show the effectiveness of our method on real datasets of varying complexity (Fashion-MNIST, Neuromophic-MNIST and Spiking Heidelberg Digits) achieving a speedup in the backward pass of up to 150x, and $85\%$ more memory efficient, without losing accuracy.

## 1   Introduction

In recent years, deep artificial neural networks (ANNs) have matched and occasionally surpassed human-level performance on increasingly difficult auditory and visual recognition problems [1, 2, 3], natural language processing tasks [4, 5] and games [6, 7, 8]. As these tasks become more challenging the neural networks required to solve them grow larger and consequently their power efficiency becomes more important [9, 10]. At the same time, the increasing interest in deploying these models into embedded applications calls for faster, more efficient and less memory intensive networks [11, 12].

The human brain manages to perform similar and even more complicated tasks while only consuming about 20W [13] which contrasts with the hundreds of watts required for running ANNs [9]. Unlike ANNs, biological neurons in the brain communicate through discrete events called spikes. A biological neuron integrates incoming spikes from other neurons in its membrane potential and after reaching a threshold emits a spike and resets its potential [14]. The spiking neuron, combined with the sparse connectivity of the brain, results in a highly spatio-temporally sparse activity [15] which is fundamental to achieve this level of energy efficiency.

Inspired by the extraordinary performance of the brain, neuromorphic computing aims to obtain the same level of energy efficiency preferably while maintaining an accuracy standard on par with ANNs by emulating spiking neural networks (SNNs) on specialised hardware [16, 17, 18, 19]. These efforts go in two directions: emulating the SNNs and training the SNNs. While there is a growing number of successful neuromorphic implementations for the former [20, 21, 22, 23, 24, 17, 25, 26], the latter has proven to be more challenging. Some neuromorphic chips implement local learning rules [26, 27, 28] and recent advances have achieved to approximate backpropagation on-chip [29]. However, a full end-to-end supervised learning via error backpropagation requires off-chip training [22, 23, 18]. This is usually achieved by simulating and training the entire SNN on a GPU and more

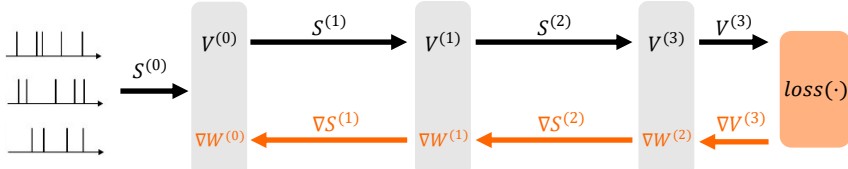

Figure 1: Spiking Neural Network diagram highlighting the forward and backward passes

recently, emulating the forward pass of the SNN on neuromorphic hardware and then performing the backward pass on a GPU [30, 31]. The great flexibility provided by GPUs allows the development of complex training pipelines without being constrained by neuromorphic hardware specifications. However, as GPUs do not leverage the event-driven nature of spiking computations this results in effective but slower and less energy efficient training. An alternative to directly training an SNN consists on training an ANN and converting it to an SNN. However, this approach has been recently shown to not be more energy efficient than using the original ANN in the first place [32].

Training SNNs has been a challenge itself even without considering a power budget due to the all-or-none nature of spikes which hinders traditional gradient descent methods [33, 34]. Since the functional form of a spike is a unit impulse, it has a zero derivative for all membrane potentials except at the threshold where it is infinity. Recently developed methods based on surrogate gradients have been shown to be capable of solving complex tasks to near state-of-the-art accuracies [35, 18, 36, 37, 38, 39, 31]. The problem of non-differentiability is solved by adopting a well-behaved surrogate gradient when backpropagating the error [33]. This method, while effective for training very accurate models, is still constrained to working on dense tensors, thus, not profiting from the sparse nature of spiking, and limits the training speed and power efficiency when training SNNs.

In this work we introduce a sparse backpropagation method for SNNs. Recent work on ANNs has shown that adaptive dropout [40] and adaptive sparsity [41] can be used to achieve state-of-the-art performance at a fraction of the computational cost [42, 43]. We show that by redefining the surrogate gradient functional form, a sparse learning rule for SNNs arises naturally as a three-factor Hebbian-like learning rule. We empirically show an improved backpropagation time up to 70x faster than current implementations and up to $40\%$ more memory efficient in real datasets (Fashion-MNIST (F-MNIST) [44], Neuromorphic-MNIST (N-MNIST) [45] and Spiking Heidelberg Dataset (SHD) [46]) thus reducing the computational gap between the forward and backward passes. This improvement will not only impact training for neuromorphic computing applications but also for computational neuroscience research involving training on SNNs [39].

## 2   Sparse Spike Backpropagation

### 2.1   Forward Model

We begin by introducing the spiking networks we are going to work with (Figs. 1 and 2). For simplicity we have omitted the batch dimension in the derivations but it is later used on the complexity analysis and experiments. We have a total of $L$ fully connected spiking layers. Each layer $l$ consisting of $N^{(l)}$ spiking neurons which are fully connected to the next layer $l+1$ through synaptic weights $W^{(l)} \in \mathbb{R}^{N^{(l)} \times N^{(l+1)}}$. Each neuron has an internal state variable, the membrane potential, $V_j^{(l)}[t] \in \mathbb{R}$ that updates every discrete time step $t \in \{0, \ldots, T-1\}$ for some finite simulation time $T \in \mathbb{N}$. Neurons emit spikes according to a spiking function $f \colon \mathbb{R} \to \{0, 1\}$ such that

$$S_i^{(l)}[t] = f(V_i^{(l)}[t]) \tag{1}$$

The membrane potential varies according to a simplified Leaky Integrate and Fire (LIF) neuron model in which input spikes are directly integrated into the membrane [14]. After a neuron spikes, its potential is reset to $V_r$. We will work with a discretised version of this model (see Appendix E for a continuous time version). The membrane potential of neuron $j$ in layer $l+1$ evolves according to the following difference equation

$$V_j^{(l+1)}[t+1] = \alpha(V_j^{(l+1)}[t] - V_{rest}) + V_{rest} + \sum_i S_i^{(l)}[t]W_{ij}^{(l)} - (V_{th} - V_r)S_j^{(l+1)}[t] \tag{2}$$

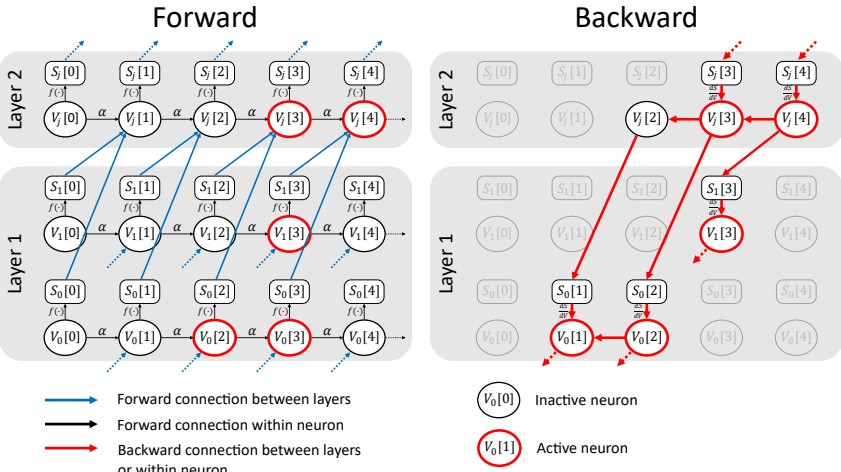

Figure 2: Illustration of the gradient backpropagating only through active neurons.

The first two terms account for the leaky part of the model where we define $\alpha = \exp(-\Delta t/\tau)$ with $\Delta t, \tau \in \mathbb{R}$ being the time resolution of the simulation and the membrane time constant of the neurons respectively. The second term deals with the integration of spikes. The last term models the resetting mechanism by subtracting the distance from the threshold to the reset potential. For simplicity and without loss of generality, we consider $V_{rest} = 0, V_{th} = 1, V_r = 0$ and $V_i^{(l)}[0] = 0 \, \forall \, l, i$ from now on.

We can unroll (2) to obtain

$$V_j^{(l+1)}[t+1] = \sum_i \underbrace{\underbrace{\sum_{k=0}^{t} \alpha^{t-k} S_i^{(l)}[k]}_{\text{Input trace}} W_{ij}^{(l)}}_{\text{Weighted input}} - \underbrace{\sum_{k=0}^{t} \alpha^{t-k} S_j^{(l+1)}[k]}_{\text{Resetting term}} \tag{3}$$

Thus, a spiking neural network can be viewed as a special type of recurrent neural network where the activation function of each neuron is the spiking function $f(\cdot)$. The spiking function, is commonly defined as the unit step function centered at a particular threshold $V_{th}$

$$f(v) = \begin{cases} 1, & v > V_{th} \\ 0, & \text{otherwise} \end{cases} \tag{4}$$

Note that while this function is easy to compute in the forward pass, its derivative, the unit impulse function, is problematic for backpropagation which has resulted in adopting surrogate derivatives [33] to allow the gradient to flow. Interestingly, it has been shown that surrogate gradient descent performs robustly for a wide range of surrogate functional forms [38].

After the final layer, a loss function $loss(\cdot)$ computes how far the network activity is with respect to some target value for the given input. The loss function can be defined as a function of the network spikes, membrane potentials or both. It may also be evaluated at every single time step or every several steps. We deliberately leave the particular definition of this function open as it does not directly affect the sparse gradient descent method we introduce.

## 2.2 Backward Model

The SNN is updated by computing the gradient of the loss function with respect to the weights in each layer. This can be achieved using backpropagation through time (BPTT) on the unrolled network. Following (1) and (3) we can derive weight gradients to be

$$\nabla W_{ij}^{(l)} = \sum_t \underbrace{\varepsilon_j^{(l+1)}[t]}_{\substack{\text{Gradient from} \\ \text{next layer}}} \overbrace{\frac{dS_j^{(l+1)}[t]}{dV_j^{(l+1)}[t]}}^{\substack{\text{Spike} \\ \text{derivative}}} \left( \underbrace{\sum_{k<t} \alpha^{t-k-1} S_i^{(l)}[k]}_{\text{Input trace}} \right) \tag{5}$$

This gradient consists of a sum over all time steps of the product of three terms. The first term, modulates the spatio-temporal credit assignment of the weights by propagating the gradient from the next layer. The second term, is the derivative of $f(\cdot)$ evaluated at a given $V_j[t]$. The last term is a filtered version of the input spikes. Notice that it is not necessary to compute this last term as this is already computed in the forward pass (3). We have chosen not make the spike resetting term in (3) differentiable as it has been shown that doing so can result in worse testing accuracy [38] and it increases the computational cost.

For a typical loss function defined on the output layer activity, the value of $\varepsilon_j^{(L-1)}[t]$ (i.e. in the last layer) is the gradient of the loss with respect to the output layer (see Fig. 1). For all other layers we have that $\varepsilon_j^{(l)}[t] = \nabla S_j^{(l)}[t]$, that is, the gradient of the loss function with respect to the output spikes of layer $l$. We can obtain an expression for $\nabla S_j^{(l)}[t]$ as a function of the gradient of the next layer:

$$
\varepsilon_i^{(l)}[t] = \nabla S_i^{(l)}[t] = \sum_j W_{ij} \left( \sum_{k>t} \overbrace{\nabla S_j^{(l+1)}[k]}^{\substack{\text{Gradient from} \\ \text{next layer}}} \overbrace{\frac{dS_j^{(l+1)}[k]}{dV_j^{(l+1)}[k]}}^{\substack{\text{Spike} \\ \text{derivative}}} \alpha^{k-t-1} \right) \tag{6}
$$
$$
= \sum_j W_{ij} \delta_j^{(l+1)}[t], \qquad l = \{0, \ldots L-2\}
$$

All hidden layers need to compute equations (5) and (6). The first one to update their input synaptic weights and the second one to backpropagate the loss to the previous layer. The first layer ($l=1$) only needs to compute (5) and the last layer ($l=L-1$) will compute a different $\varepsilon_j^{(L-1)}[t]$ depending on the loss function.

We introduce the following definitions which result in the sparse backpropagation learning rule.

**Definition 1.** *Given a backpropagation threshold $B_{th} \in \mathbb{R}$ we say neuron $j$ is **active** at time $t$ iff*

$$
|V_j[t] - V_{th}| < B_{th} \tag{7}
$$

**Definition 2.** *The spike gradient is defined as*

$$
\frac{dS_j[t]}{dV_j[t]} := \begin{cases} g(V_j[t]), & \text{if } V_j[t] \text{ is active} \\ 0, & \text{otherwise} \end{cases} \tag{8}
$$

This means that neurons are only active when their potential is close to the threshold. Applying these two definitions to (5) and (6) results in the gradients only backpropagating through active neurons at each time step as shown in Fig. 2. The consequences of this are readily available for the weight gradient in (5) as the only terms in the sum that will need to be computed are those in which the postsynaptic neuron $j$ was active at time $t$. Resulting in the following gradient update:

$$
\nabla W_{ij}^{(l)}[t] = \begin{cases} \varepsilon_j^{(l+1)}[t] \frac{dS_j^{(l+1)}[t]}{dV_j^{(l+1)}[t]} \left( \sum_{k<t} \alpha^{t-k-1} S_i^{(l)}[t] \right) & V_j^{(l+1)}[t] \text{ is active,} \\ 0, & \text{otherwise} \end{cases} \tag{9}
$$
$$
\nabla W_{ij}^{(l)} = \sum_t \nabla W_{ij}^{(l)}[t] \tag{10}
$$

For the spike gradient in (6) we have two consequences, firstly, we will only need to compute $\nabla S_j^{(l)}[t]$ for active neurons since $\nabla S_j^{(l)}[t]$ is always multiplied by $\frac{dS_j^{(l)}[t]}{dV_j^{(l)}[t]}$ in (5) and (6). Secondly, we can use a recurrent relation to save time and memory when computing $\delta_j[t]$ in (6) as shown in the following proposition.

**Proposition 1.** *We can use a recurrent relation to compute $\delta_j[t]$ given by*

$$\delta_j[t] = \begin{cases} \alpha^n \delta_j[t+n], & \text{if } \frac{dS_j[k]}{dV_j[k]} = 0, \text{ for } t+1 \leq k \leq t+n \\ \nabla S_j[t+1] \frac{dS_j[t+1]}{dV_j[t+1]} + \alpha^n \delta_j[t+n], & \text{if } \frac{dS_j[k]}{dV_j[k]} = 0, \text{ for } t+1 < k \leq t+n, \frac{dS_j[t+1]}{dV_j[t+1]} \neq 0 \end{cases} \tag{11}$$

This means that we only need to compute $\delta_j^{(l+1)}[t]$ at those time steps in which either $\frac{dS_j[t+1]}{dV_j[t+1]} \neq 0$ or $V_j^{(l)}[t]$ is active (see Appendix C for a visualisation of this computation). Thus, we end up with the following sparse expression for computing the spike gradient.

$$\nabla S_j^{(l)}[t] = \begin{cases} \sum_j W_{ij} \delta_j^{(l+1)}[t], & V_j^{(l)}[t] \text{ is active} \\ 0, & \text{otherwise} \end{cases} \tag{12}$$

## 2.3 Complexity analysis

We define $\rho_l \in [0,1]$ to be the probability that a neuron is active in layer $l$ at a given time. Table 1 summarises the computational complexity of the gradient computation in terms of number of sums and products. We use here $N$ to refer to the number of neurons in either the input or output layer and $B$ to refer to the batch size. In a slight abuse of notation we include the constants $\rho_l$ as part of the complexity expressions. Details of how these expression were obtained can be found in Appendix B and are based on a reasonable efficient algorithm that uses memoisation for computing $\delta_j[t]$.

Table 1: Computational complexity

|  | Original | Sparse |
|---|---|---|
| Sums $\nabla W^{(l)}$ | $O(BTN^2)$ | $O(\rho_{l+1}BTN^2)$ |
| Products $\nabla W^{(l)}$ | $O(BTN^2)$ | $O(\rho_{l+1}BTN^2)$ |
| Sums $\nabla S^{(l)}$ | $O(BTN^2)$ | $O(\rho_l BTN^2))$ |
| Products $\nabla S^{(l)}$ | $O(BTN^2)$ | $O(\rho_l BTN^2)$ |

In order for the sparse gradients to work better than their dense counterpart we need to have a balance between having few enough active neurons as to make the sparse backpropagation efficient while at the same time keeping enough neurons active to allow the gradient to backpropagate. We later show in section 3 that this is the case when testing it on real world datasets.

## 2.4 Sparse learning rule interpretation

The surrogate derivative we propose, while not having a constrained functional form for active neurons, imposes a zero gradient for the inactive ones. Looking back into equation (5) we can identify it now as a three-factor Hebbian learning rule [47]. With our surrogate derivative definition, the second term (spike derivative) measures whether the postsynaptic neuron is close to spiking (i.e. its membrane potential is close to the threshold). The third term (input trace) measures the presynaptic neuron activity in the most recent timesteps. Finally the first term (gradient from next layer) decides whether the synaptic activity improves or hinders performance with respect to some loss function. We note that this is not exactly a Hebbian learning rule as neurons that get close to spike but do not spike can influence the synaptic weight update and as such, the second term is simply a surrogate of the postsynaptic spike activity.

Previous surrogate gradient descent methods have used a similar approach to implement backpropagation on SNNs [18, 36, 48, 39, 38], often using surrogate gradient descent definitions that are maximal when the membrane is at the threshold and decay as the potential moves away from it. In fact, in Esser et al. [18] the surrogate gradient definition given implies that neurons backpropagate a zero gradient 32% of the time. However, as shown in the next section, neurons can be active a lot less often when presented with spikes generated from real world data and still give the same test accuracy. This is only possible because surrogate gradient approximations to the threshold function yield much larger values when the potential is closer to spike thus concentrating most of the gradient on active neurons. This fact makes it possible to fully profit from our sparse learning rule for a much reduced computational and memory cost.

# 3 Experiments

We evaluated the correctness and improved efficiency of the proposed sparse backpropagation algorithm on real data of varying difficulty and increasing spatio-temporal complexity. Firstly, the Fashion-MNIST dataset (F-MNIST) [44] is an image dataset that has been previously used on SNNs by converting each pixel analogue value to spike latencies such that a higher intensity results in an earlier spike [39][38]. Importantly, each input neuron can only spike at most once thus resulting in a simple spatio-temporal complexity and very sparse coding. Secondly, we used the Neuromorphic-MNIST (N-MNIST) [45] dataset where spikes are generated by presenting images of the MNIST dataset to a neuromorphic vision sensor. This dataset presents a higher temporal complexity and lower sparsity than the F-MNIST as each neuron can spike several times and there is noise from the recording device. Finally, we worked on the Spiking Heidelberg Dataset (SHD) [46]). This is a highly complex dataset that was generated by converting spoken digits into spike times by using a detailed model of auditory bushy cells in the cochlear nucleus. This results in a very high spatio-temporal complexity and lowest input sparsity. A visualisation of samples of each dataset can be found in the supplementary materials. These datasets cover the most relevant applications of neuromorphic systems. Namely, artificially encoding a dataset into spikes (Fashion-MNIST), reading spikes from a neuromorphic sensor such as a DVS camera (N-MNIST) and encoding complex temporally-varying data into spikes (SHD).

We run all experiments on three-layer fully connected network as in Fig. 1, where the last layer (readout) has an infinite threshold. Both spiking layers in the middle have the same number of neurons. We used $g(V) := 1/(\beta|V - V_{th}| + 1)^2$ for the surrogate gradient as in [38]. See Appendix E for all training details. We implemented the sparse gradient computation as a Pytorch CUDA extension [49].

## 3.1 Spiking neurons are inactive most of the time resulting in higher energy efficiency

One of the most determining factors of the success of our sparse gradient descent resides on the level of sparsity that we can expect and, consequently, the proportion of active neurons we have in each layer. We measure the percentage of active neurons on each batch as training progresses for each dataset. This can be computed as

$$\text{Activity} = 100 \times \frac{\text{\# of active neurons}}{BTN} \tag{13}$$

where $B$ is the batch size $T$ is the total number of time steps and $N$ the number of neurons in the layer and the number of active neurons is obtained according to (7). This is an empirical way of measuring the coefficient $\rho_l$ introduced earlier.

Figure 3A shows the activity on each dataset. As expected, the level of activity is correlated with the sparseness of the inputs with the lowest one being on the F-MNIST and the larger in the SHD dataset. Remarkably however, the activity on average is never above 2% on average in any dataset and it is as low as 1.06% in the F-MNIST dataset. This means that on average we will never have to compute more than 2% of the factors in $\nabla W$ and 2% of the values of $\nabla S$. This can be visualised in the bottom row of Fig. 3B.

These coefficients give us a theoretical upper bound to the amount of energy that can be saved if this sparse backpropagation rule was implemented in specialised hardware which only performed the required sums and products. This is summarised in Table 2.

Table 2: Theoretical upper bound to energy saved in hidden layers.

|  | Mean activity layer 1 (%) | Mean activity layer 2 (%) | Energy saved in $\nabla W$ (%) | Energy saved in $\nabla S$ (%) |
|---|---|---|---|---|
| F-MNIST | 1.06 | 0.87 | 99.13 | 98.94 |
| N-MNIST | 1.12 | 0.77 | 99.23 | 98.88 |
| SHD | 1.70 | 1.09 | 98.91 | 98.30 |

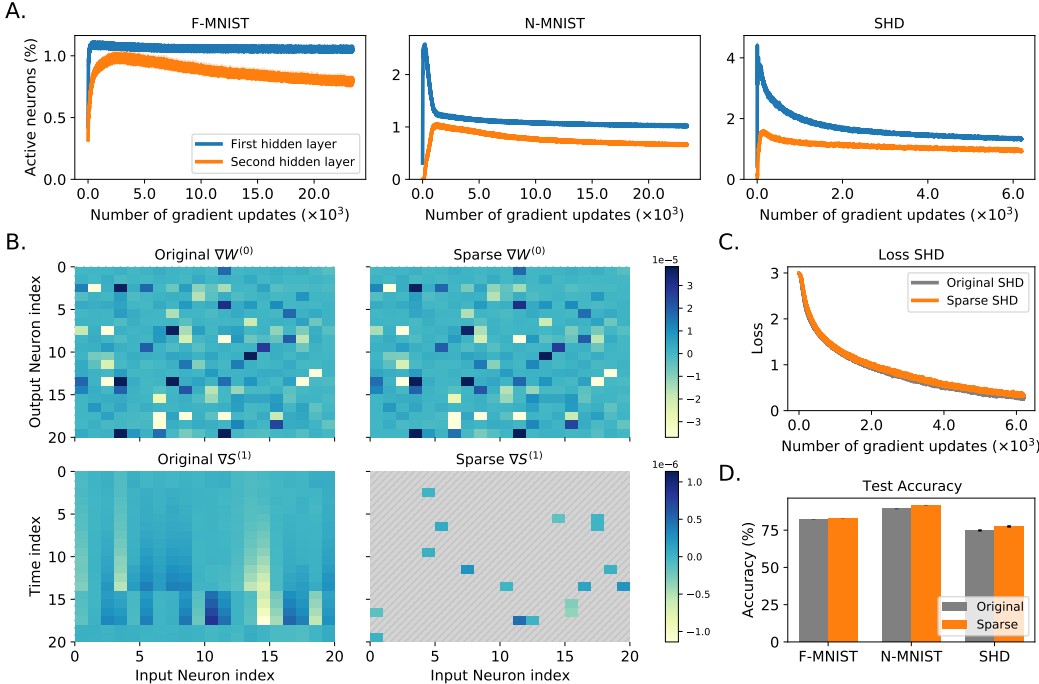

Figure 3: Sparse backpropagation learns at high levels of sparsity. All figures except B are a 5 sample average. Standard error in the mean is displayed (although too small to be easily visualised). **A.** Percentage of active neurons during training as a fraction of total tensor size ($B \times T \times N$) with $N = 200$ neurons for each dataset. **B.** Visualisation of the weight and spike gradients on the SHD dataset. We show zero value in hatched grey. Note how both $\nabla W^{(0)}$ are nearly identical in despite being computed using a small fraction of the values of $\nabla S^{(1)}$ in the sparse case. **C.** Loss evolution on the SHD dataset using both algorithms. **D.** Final test accuracy on all datasets using both methods.

Importantly, in dense backpropagation methods the temporal dimension of datasets and models is necessarily limited to a maximum of $\sim 2000$ since gradients must be computed at all steps [46, 39]. This means that a lot of negligible gradients are computed. However, our method adapts to the level of activity of the network and only computes non-negligible gradients. This cost-prohibitive reality was evidenced when we attempted to run these experiments in smaller GPUs leading to running out of memory when using dense methods but not on ours (see F.8). Thus, sparse spiking backpropagation allows to train data that runs for longer periods of time without requiring a more powerful hardware.

### 3.2 Sparse backpropagation training approximates the gradient very well

We test that our sparse learning rule performs on par with the standard surrogate gradient descent. Figure 3B shows a visualisation of the weight (first row) and spike gradients (second row). Note how most of the values of $\nabla S^{(1)}$ were not computed and yet $\nabla W^{(1)}$ is nearly identical to the original gradient. This is further confirmed in Figures 3C and 3D where the loss on the SHD dataset is practically identical with both methods (the loss for the other datasets can be found in Appendix F) and the test accuracy is practically identical (albeit slightly better) in the sparse case (F-MNIST: $82.2\%$, N-MNIST: $92.7\%$, SHD: $77.5\%$). These accuracies are on par to those obtained with similar networks on these datasets (F-MNIST: $80.1\%$, N-MNIST: $97.4\%$, SHD: $71.7\%$) [46, 38, 39]. These results show that sparse spiking backpropagation gives practically identical results to the original backpropagation but at much lower computational cost.

### 3.3 Sparse training is faster and less memory intensive

We now measure the time it takes the forward and the backward propagation of the second layer of the network during training as well as the peak GPU memory used. We choose this layer because it

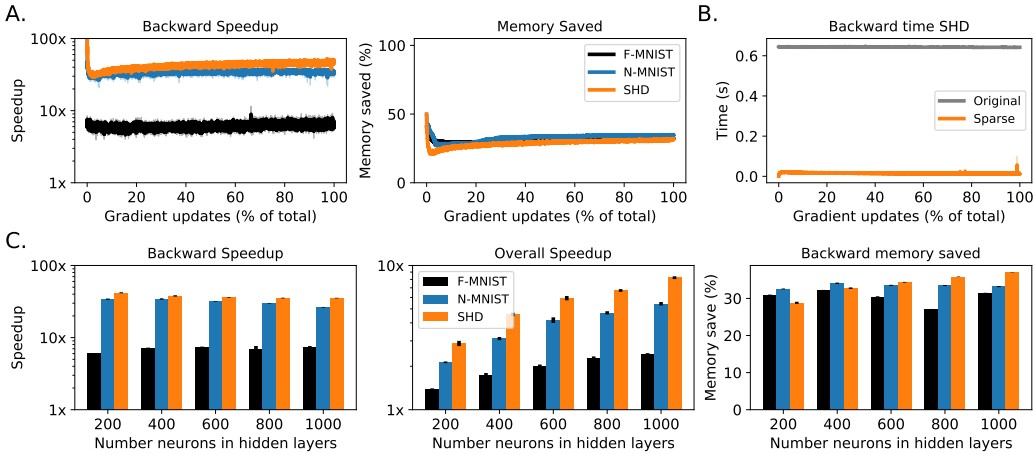

Figure 4: Speedup and memory improvement when using sparse gradient backpropagation. All figures are a 5 sample average and its standard error in the mean is also displayed. **A.** Backward speedup and memory saved in the second hidden layer for all three datasets when using sparse gradient descent with 200 hidden neurons. **B.** Time spent in computing the backward pass in the second hidden layer consisting of 200 neurons when using regular gradient descent and sparse gradient descent. **C.** Backward speed and memory saved in the second hidden layer as we increase the number of hidden neurons in all hidden layers. We included the overall speedup which takes into consideration both the forward and backward times spent in the second layer.

needs to compute both $\nabla W^{(1)}$ and $\nabla S^{(1)}$ (note that the first layer only needs to compute $\nabla W^{(0)}$). A performance improvement in this layer translates to an improvement in all subsequent layers inn a deeper network. We compute the backward speedup as the ratio of the original backpropagation time and the sparse backpropagation time. We compute the memory saved as

$$\text{Memory saved} = 100 \times \frac{\text{memory\_original} - \text{memory\_sparse}}{\text{memory\_original}} \tag{14}$$

Figure 4A shows the speedup and memory saved on a 200 neuron hidden layer (both hidden layers). Backpropagation time is faster by $40x$ and GPU memory is reduced by $35\%$. To put this speedup into perspective Figure 4B shows the time taken for backpropagating the second layer in the original case (about $640$ms) and the sparse case (about $17$ms). It also shows that the time taken for backpropagating is nearly constant during training as it could be expected after studying the network activity in Fig. 3A.

We also tested how robust is this implementation for a varying number of hidden neurons (in both layers). We increase the number of neurons up to $1000$ which is similar to the maximum number of hidden units used in previous work [46] and it is also the point at which the original backpropagation method starts to run out of memory. Here we also show the overall speedup defined as the ratio between the sum of forward and backward times of the original over the sparse backpropagation. The results shown in Fig. 4C show that the backward speedup and memory saved remain constant as we increase the number of hidden neurons but the overall speedup increases substantially. Given that the forward time is the same for both methods (see Appendix F) this shows that as we increase the number of neurons, the backward time takes most of the computation and thus the sparse backpropagation becomes more important.

We also note that these results vary depending on the GPU used, Figure 4 was obtained from running on an RTX6000 GPU. We also run this on smaller GPUs (GTX1060 and GTX1080Ti) and found that the improvement is even better reaching up to a 70x faster backward pass on N-MNIST and SHD. These results can found in Appendix F. Our results show that our method speeds up backward execution between one and two orders of magnitude, meaning that to a first approximation we would expect it reduce energy usage by the same factor since GPUs continue to use 30-50% of their peak power even when idle [50].

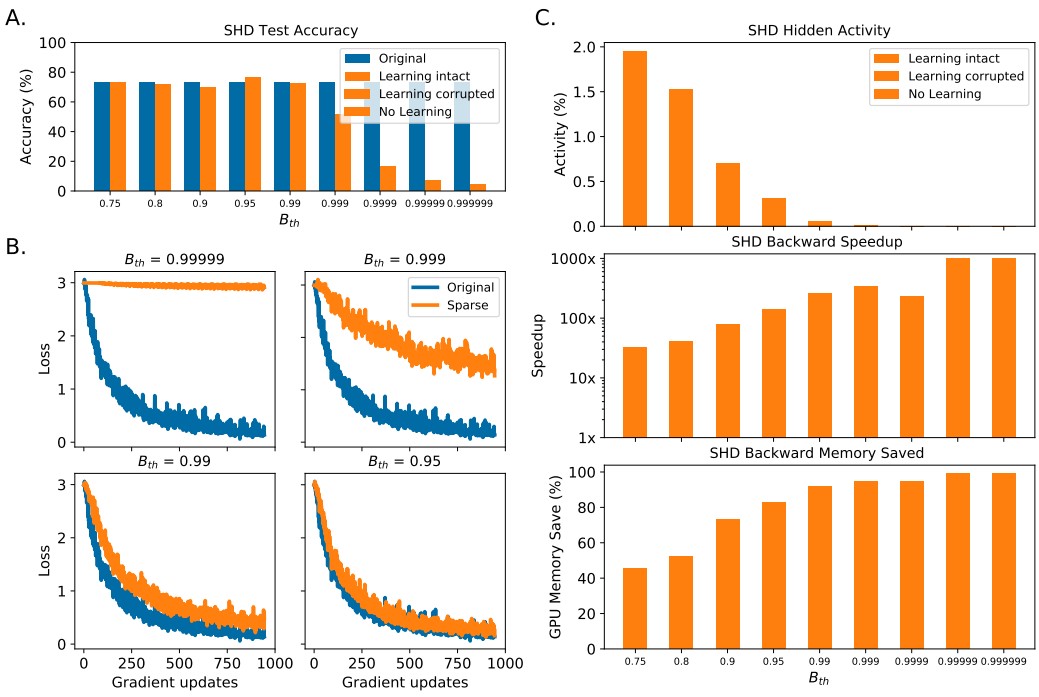

Figure 5: Impact of varying the backpropagation threshold $B_{th}$ on the SHD dataset. **A.** Final test accuracy on the SHD dataset as $B_{th}$ increases. **B.** Loss evolution on the SHD dataset for different $B_{th}$. **C.** Hidden activity, backward speedup and backward memory saved as $B_{th}$ increases.

### 3.4 Training deeper and more complex architectures

We show that our gradient approximation can successfully train deeper networks by training a 6-layer fully connected network (5 hidden plus readout all with 300 neurons) on Fashion-MNIST. We measured the performance of the 5 hidden layers and achieve a 15x backward speedup when using 1080-Ti GPU (better than the 10x speedup under these conditions with only 2 hidden layers as shown in Appendix F.8) and $25\%$ memory saved while achieving a $82.7\%$ testing accuracy (slightly better than with 2 layers at $81.5\%$). The results are consistent with our previous findings and show that the gradient is not loss in deeper architectures.

We also run a experiment with a convolutional and pooling layers on the Fashion-MNIST dataset obtaining a test accuracy of $86.7\%$ when using our spike gradient approximation and $86.9\%$ with the original gradients. We do not report speedup or memory improvements in this network as developing a sparse convolutional CUDA kernel is out of the scope of our work but we simply ran a dense implementation with clamped gradients. This proves that our gradient approximation is able to train on more complex architectures although it still remains to be shown whether efficient sparse operators can be implemented for this purpose. See Appendix E for training details.

### 3.5 Sparsity-Accuracy trade-off

We trained the SHD dataset on the original 2-layer fully connected network with $400$ neurons in each hidden layer and varied the backpropagation threshold $B_{th}$. We found that the training is very robust to even extreme values of $B_{th}$. The loss and accuracy remains unchanged even when setting $B_{th} = 0.95$ (see 5A and 5B) achieving a nearly $150$x speedup and $85\%$ memory improvement as shown in 5C. We also inspected the levels of activity while varying $B_{th}$ and we found that there are enough active neuron to propagate the gradient effectively. As $B_{th}$ gets closer to 1 the number of active neurons decreases rapidly until there are no gradients to propagate.

# 4    Discussion

Recent interest in neuromorphic computing has lead to energy efficient emulation of SNNs in specialised hardware. Most of these systems only support working with fixed network parameters thus requiring an external processor to train. Efforts to include training within the neuromorphic system usually rely on local learning rules which do not guarantee state-of-the-art performance. Thus, training on these systems requires error backpropagation on von-Neumann processors instead of taking advantage of the physics of the neuromorphic substrate [51, 52].

We have developed a sparse backpropagation algorithm for SNNs that achieves the same performance as standard dense backpropagation at a much reduced computational and memory cost. To the best of our knowledge this is the first backpropagation method that leverages the sparse nature of spikes in training and tackles the main performance bottleneck in neuromorphic training. Moreover, while we have used dense all-to-all connectivity between layers, our algorithm is compatible with other SNNs techniques that aim for a sparser connectivity such as synaptic pruning [53].

By constraining backpropagation to active neurons exclusively, we reduce the number of nodes and edges in the backward computation graph to a point where over $98\%$ of the operations required to compute the gradients can be skipped. This results in a much faster backpropagation time, lower memory requirement and less energy consumption on GPU hardware while not affecting the testing accuracy. This is possible because the surrogate gradient used to approximate the derivative of the spiking function is larger when the membrane potential is closer to the threshold resulting in most of the gradient being concentrated on active neurons. Previous work has used, to a lesser extent, similar ideas. Esser et al. [18] used a surrogate gradient function which effectively resulted in about one third the neurons backpropagating a zero gradient. Later, Pozzi, et al. [54] developed a learning rule where only the weights of neurons directly affecting the activity of the selected output class are updated, however, the trial and error nature of the learning procedure resulted in up to $3.5x$ slower training.

However, due to the lack of efficient sparse operators in most auto-differentiation platforms, every layer type requires to develop its own sparse CUDA kernel in order to be competitive with current heavily optimised libraries. In our work, we concentrated on implementing and testing this in spiking fully connected layers and we managed to show that sparse backpropagation is a lot faster and uses less memory. Developing these sparse kernels is not trivial and we are aware that having to do so for each layer is an important limitation of our work. This is no surprise since for now there is few reasons to believe that sparse tensor operations will significantly accelerate neural network performance. Our work aims to challenge this view and motivate the adoption of efficient sparse routines by showing for the first time that sparse BPTT can be faster without losing accuracy.

We implemented and tested our method on real data on fully connected SNNs and simulated it in convolutional SNNs. The same method can be extended to any SNN topology as long as the neurons use a threshold activation function and the input is spiking data. However, due to the lack of efficient sparse operators in most auto-differentiation platforms, every layer type requires to develop its own sparse CUDA kernel in order to be competitive with current heavily optimised libraries. Developing these sparse kernels is not trivial and we are aware that having to do so for each layer is an important limitation of our work. This is no surprise since for now there is few reasons to believe that sparse tensor operations will significantly accelerate neural network performance. Our work aims to challenge this view and motivate the adoption of efficient sparse routines by showing for the first time that sparse spiking BPTT can be faster while achieving the same accuracy as its dense counterpart. Nevertheless, ideally, a dynamic backward graph should be generated each time the backward pass needs to be computed and consequently a GPU may not be the best processing unit for this task. A specialised neuromorphic implementation or an FPGA may be more suited to carry out the gradient computation task as this graph changes every batch update. Additionally, while we have reduced the number of operations required for training several times, we have only achieved up to a $85\%$ memory reduction thus, memory requirements remains an important bottleneck.

In summary, our sparse backpropagation algorithm for spiking neural networks tackles the main bottleneck in training SNNs and opens the possibility of a fast, energy and memory efficient end-to-end training on spiking neural networks which will benefit neuromorphic training and computational neuroscience research. Sparse spike backpropagation is a necessary step towards a fully efficient on-chip training which will be ultimately required to achieve the full potential of neuromorphic computing.

## Acknowledgments and Disclosure of Funding

We thank Friedemann Zenke, Thomas Nowotny and James Knight for their feedback and support. This work was supported by the EPSRC Centre for Doctoral Training in High Performance Embedded and DistributedSystems (HiPEDS, Grant Reference EP/L016796/1)

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
