# Sparse Spiking Gradient Descent: Supplementary Materials

## A Proof of Proposition 1

**Proposition 1.** *We can use a recurrent relation to compute $\delta_j[t]$ given by*

$$\delta_j[t] = \begin{cases} \alpha^n \delta_j[t+n], & \text{if } \frac{dS_j[k]}{dV_j[k]} = 0, \text{ for } t+1 \le k \le t+n \\ \nabla S_j[t+1]\frac{dS_j[t+1]}{dV_j[t+1]} + \alpha^n \delta_j[t+n], & \text{if } \frac{dS_j[k]}{dV_j[k]} = 0, \text{ for } t+1 < k \le t+n, \frac{dS_j[t+1]}{dV_j[t+1]} \ne 0 \end{cases} \tag{15}$$

**Proof.** *The proof follows directly from the definition of $\delta_j[t]$.*

*For the first case we have*

$$\delta_j[t] = \sum_{k>t} \nabla S_j[k]\frac{dS_j[k]}{dV_j[k]}\alpha^{k-t-1}$$

$$= \nabla S_j[t+1]\frac{dS_j[t+1]}{dV_j[t+1]}\alpha^0 + S_j[t+2]\frac{dS_j[t+2]}{dV_j[t+2]}\alpha^1 + \ldots + \nabla S_j[t+n]\frac{dS_j[t+n]}{dV_j[t+n]}\alpha^{n-1}$$

$$+ \nabla S_j[t+n+1]\frac{dS_j[t+n+1]}{dV_j[t+n+1]}\alpha^n + \ldots$$

$$= \nabla S_j[t+1]\frac{dS_j[t+1]}{dV_j[t+1]}\alpha^0 + S_j[t+2]\frac{dS_j[t+2]}{dV_j[t+2]}\alpha^1 + \ldots + \nabla S_j[t+n]\frac{dS_j[t+n]}{dV_j[t+n]}\alpha^{n-1}$$

$$+ \sum_{k>t+n} \nabla S_j[k]\frac{dS_j[k]}{dV_j[k]}\alpha^{k-t-1}$$

$$= \alpha^n \sum_{k>t+n} \nabla S_j[k]\frac{dS_j[k]}{dV_j[k]}\alpha^{k-(t+n)-1}$$

$$= \alpha^n \delta_j[t+n]$$

*where the penultimate equality holds from the condition $\frac{dS_j[k]}{dV_j[k]} = 0$, for $t+1 \le k \le t+n$*

*The proof of the second case is identical except that now the term $\nabla S_j[t+1]\frac{dS_j[t+1]}{dV_j[t+1]}$ will be non-zero.*

## B Complexity analysis

For clarity, we consider that all layers have the same number $N$ of neurons. The weight gradient is given by

$$\nabla W_{ij}^{(l)} = \sum_t \nabla S_j^{(l+1)}[t]\frac{dS_j^{(l+1)}[t]}{dV_j^{(l+1)}[t]}\left(\sum_{k<t} \alpha^{t-k-1} S_i^{(l)}[t]\right)$$

Since the trace $\left(\sum_{k<t} \alpha^{t-k-1} S_i^{(l)}[t]\right)$ has been compated in the forward pass, then computing $\nabla W_{ij}^{(l)}$ requires doing a total of $T$ products and $T-1$ sums per element in $W$ per batch. Thus resulting in $O(BTN^2)$ products and $O(BTN^2)$ sums. In the sparse case instead of $T$ we have on average $\rho_{l+1}T$ products and $\rho_{l+1}(T-1)$ sums resulting in a total of $O(\rho_{l+1}BTN^2)$ products and $O(\rho_{l+1}BTN^2)$ sums.

The spikes gradient is given by

$$\nabla S_i^{(l)}[t] = \sum_j W_{ij} \left( \sum_{k>t} \nabla S_j^{(l+1)}[t] \frac{dS_j^{(l+1)}[t]}{dV_j^{(l+1)}[t]} \alpha^{k-t-1} \right)$$
$$= \sum_j W_{ij} \delta_j^{(l+1)}[t]$$

This gradient is obtained by doing the matrix product between $W \in \mathbb{R}^{N \times N}$ and $\delta^{(l+1)} \in \mathbb{R}^{N \times T}$ per batch. This gives a total of $O(BN^2T)$ products and sums. In the sparse case we only need to compute a fraction $\rho_l$ of the entries in $\nabla S^{(l)}$ thus a total of $O(\rho_l BN^2T)$ products and sums.

There are two main ways of computing $\delta^{(l+1)}$. The naive way requires doing $O(NT^2)$ products and $O(NT^2)$ sums by computing all values with $k > t$ for a given $t$ every time. Alternatively, we can use memoisation to store all values already computed resulting in just $O(NT)$ products and $O(NT)$ sums per batch. Thus resulting in $O(BNT)$ products and sums. In the sparse case, the worst case scenario is when the active times of layers $l$ and $l+1$ are completely disjoint as we show in Appendix C. Meaning we have to compute $O((\rho_l + \rho_{l+1})BNT^2)$ products and sums without memoisation or $O((\rho_l + \rho_{l+1})BNT)$ with memoisation.

Overall we get that if we do not use memoisation for computing $\nabla S^{(l)}$ then we do $O(B(NT^2 + N^2T))$ sums and products and $O(B((\rho_{l+1} + \rho_l)NT^2 + \rho_{l+1}N^2T)$ in the sparse case. With memoisation we do $O(BN^2T)$ and $O(\rho_{l+1}BN^2T)$ in the sparse case.

## C   Visualisation of $\delta_j[t]$ computation

Given a presynaptic neuron $i$ and postsynaptic neuron $j$ we need to compute the gradient

$$\nabla S_i[t] = \sum_j W_{ij} \left( \sum_{k>t} \nabla S_j[t] \frac{dS_j[t]}{dV_j[t]} \alpha^{k-t-1} \right)$$
$$= \sum_j W_{ij} \delta_j[t]$$

We first note that $\nabla S_i[t]$ is always multiplied by $\frac{dS_i[t]}{dV_i[t]}$ either in the weight gradient equation (5) or in the spike gradient equation (6). Thus, we only need $\nabla S_i[t]$ at those times at which $\frac{dS_i[t]}{dV_i[t]} \neq 0$.

Secondly, according to Proposition 1 $\delta_j[t]$ is only modified beyond an attenuation factor of $\alpha^n$ at those times when $\frac{dS_j[t]}{dV_j[t]} \neq 0$.

We define

$$t_i = \{t : \frac{dS_i[t]}{dV_i[t]} \neq 0\} \quad \text{The times we } \textit{need} \text{ to compute the gradient of the presynaptic neuron } i$$

$$t_j = \{t : \frac{dS_j[t]}{dV_j[t]} \neq 0\} \quad \text{The times that affect } \delta_j \text{ beyond } \alpha^n \text{ attenuation}$$

Both sets of times are known by the time we compute the gradients since they are simply the times at which neurons $i$ and $j$ were active.

The following example shows how the computation of $\delta_j[t]$ would take place using Proposition 1. We note that we only need to write the result in global memory at times $t_i$. Consequently, once we have all $t_i$ we can simply stop the computation

| $t$ | 0 | 1 | 2 | 3 | 4 | 5 | 6 | 7 | 8 | 9 |
|---|---|---|---|---|---|---|---|---|---|---|
| $t_j$ | | 1 | | 3 | | | 6 | | 8 | |
| $t_i$ | | | | 3 | | 5 | 6 | | | |

$$\delta_j[7] = \frac{dS_j[8]}{dV_j[8]} \qquad \text{Compute}$$

$$\delta_j[6] = \alpha\delta_j[7] \qquad \text{Compute and write}$$

$$\delta_j[5] = \frac{dS_j[6]}{dV_j[6]} + \alpha\delta_j[6] \qquad \text{Compute and write}$$

$$\delta_j[3] = \alpha^2\delta_j[5] \qquad \text{Compute and write}$$

$$\delta_j[2] = \frac{dS_j[3]}{dV_j[3]} + \alpha\delta_j[3] \qquad \text{Not required}$$

$$\delta_j[0] = \frac{dS_j[1]}{dV_j[1]} + \alpha^2\delta_j[2] \qquad \text{Not required}$$

Thus, in the worse case scenario where sets $t_i$ and $t_j$ are disjoint instead of doing $T$ updates we only do $|t_i| + |t_j|$ updates.

## D  Visualisation of input data

Visualisation of several of one sample per dataset to provide and intuitive idea of the level of sparsity of the data.

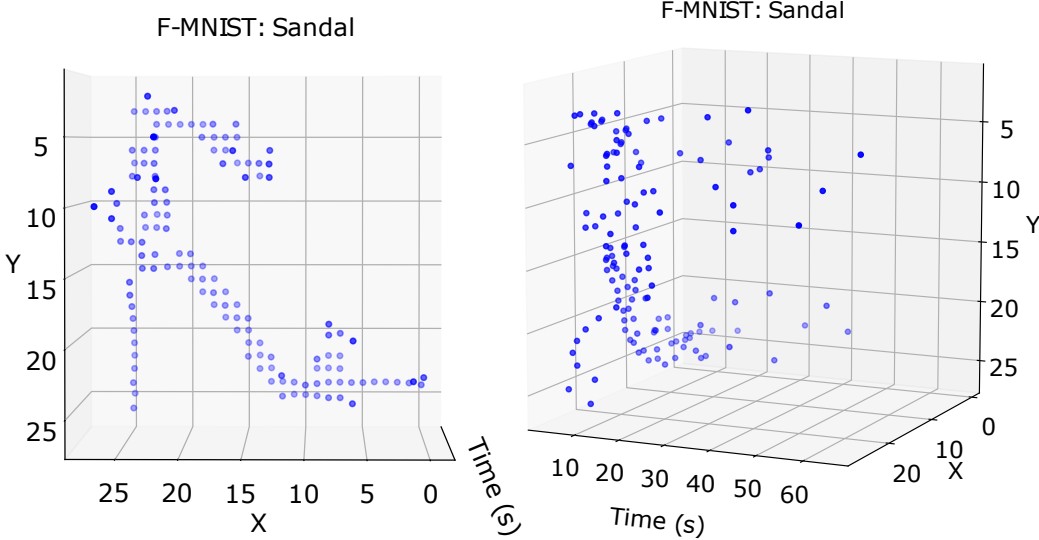

Figure 6: Visulization of a sample of the Fashion-MNIST dataset converted into spike times. Left: Front view. Right: Side view

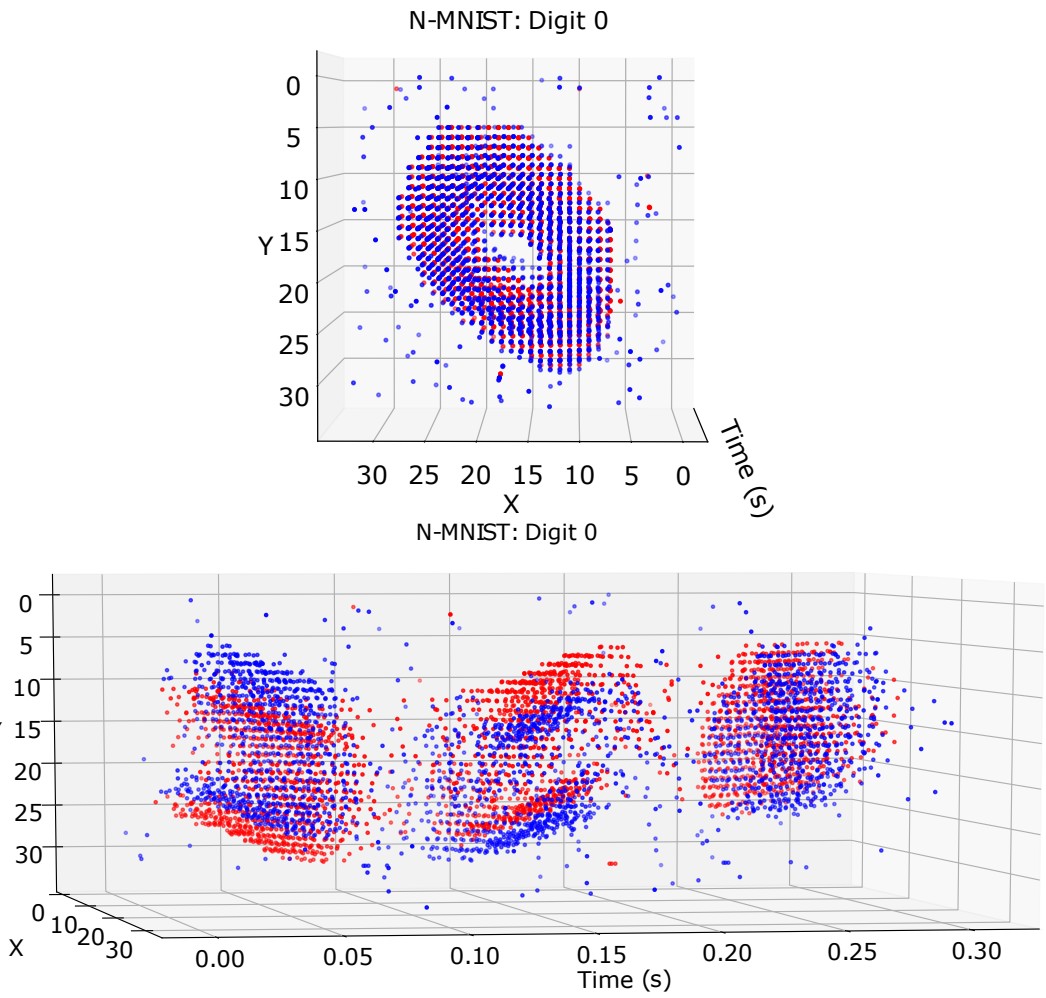

Figure 7: Visulization of a sample of the Neuromorphic-MNIST dataset. Top: Front view. Bottom: Side view

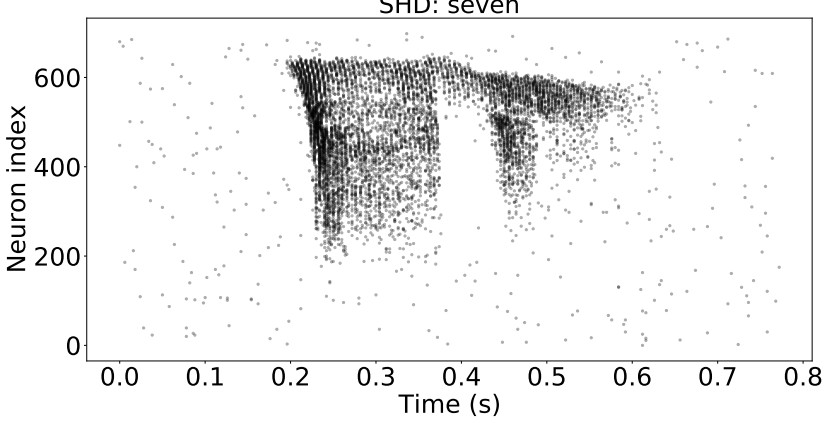

Figure 8: Visulization of a sample of the Spiking Heidelberg Digits dataset.

# E Training details

## E.1 Conversion of Fashion-MNIST to spike latencies

We convert the normalised analog grayscale pixel values $x_i$ of the Fashion-MNIST samples [44] to spike times in a similar way to that used in [38]. Specifically, we convert the normalised intensities into spike times following

$$T(x) = \begin{cases} \tau_{\text{eff}} \log\left(\frac{x}{x-\theta}\right), & x > \theta \\ \infty, & \text{otherwise} \end{cases}$$

## E.2 Spiking neuron model

We use the simplified Leaky Integrate and Fire model which is a variation on the standard Leaky Integrate and Fire model in which spikes are directly integrated in the membrane without being filtered by beforehand. The membrane potential evolves according to

$$\tau \frac{dV_j^{(l+1)}(t)}{dt} = -(V_j^{(l+1)}(t) - V_{rest}) + \tau \sum_i S_i^{(l)}[t]W_{ij}^{(l)} \tag{16}$$

The membrane potential resets to $V_r$ as soon as it reaches a threshold $V_{th}$. The model keeps the main features of a spiking model, namely, a membrane potential that varies according to the incoming input spikes and synaptic weights, a leaky component that draws the potential towards $V_{rest}$ and spiking and resetting mechanisms.

We use a discretised version of the model as in Equation (2) that we repeat here

$$V_j^{(l+1)}[t+1] = -\alpha(V_j^{(l+1)}[t] - V_{rest}) + \sum_i S_i^{(l)}[t]W_{ij}^{(l)} - (V_{th} - V_r)S_j^{(l+1)}[t] \tag{17}$$

where we define $\alpha = exp(-\Delta t/\tau)$. Spiking then takes place by applying $f(\cdot)$ on the membrane potential

$$S_j^{(l+1)}[t+1] = f(V_j^{(l+1)}[t+1]) \tag{18}$$

which is defined as a unit step function centered at the threshold $V_{th}$

$$f(v) = \begin{cases} 1, & v > V_{th} \\ 0, & \text{otherwise} \end{cases} \tag{19}$$

Thus, given an input spikes tensor $S^{(l)} \in \{0,1\}^{B \times T \times N^{(l)}}$ ($B$ being the batch size $T$ number of time steps and $N^{(l)}$ number of neurons in layer $l$) and weights $W^{(l)} \in \mathbb{R}^{N^{(l)} \times N^{(l+1)}}$ we obtain the output spikes $S^{(l+1)} \in \{0,1\}^{B \times T \times N^{(l+1)}}$ by applying (17), (18), (19).

## E.3 Network and weight initialisation

All layers are feed-forward fully connected in all experiments unless otherwise specified. The weights were sampled from a uniform distribution $\mathcal{U}(-\sqrt{N}, \sqrt{N})$ where $N$ is the number of input connections to the layer. In all experiments we have an input layer, two hidden layers with the same number of neurons and a readout layer with as many neurons as output classes for the given dataset. The readout layer neurons are identical to the hidden layers except in the firing threshold which we set to infinity.

In the convolutional layer experiment, the first layer was a convolutional spiking layer with 64 filters, kernel size of 3 and stride of 1, followed by a max pool layer with kernel size of 2 and then flattened into a linear layer of 12544 inputs and 10 outputs.

### E.4 Supervised and regularisation losses

We have a loss composed of three terms: cross entropy loss, higher activity regularisation and lower activity regularisation. The cross entropy loss is defined as

$$\mathcal{L}^{cross} = \frac{1}{B} \sum_{b=1}^{B} \sum_{c=1}^{C} y_{b,c} \log(p_{b,c})$$

With $y_{b,c} \in \{0,1\}^C$ being a one hot target vector for batch sample $b$ given a total batch size $B$ and a total number of classes $C$. The probabilities $p_{b,c}$ are obtained using the Softmax function

$$p_{b,c} = \frac{e^{a_{b,c}}}{\sum_{c=1}^{C} e^{a_{b,c}}}$$

Here, the logits $a_{b,c} \in \mathbb{R}^{B \times C}$ are obtained by taking the max over the time dimension on the readout layer membrane potential $a_{b,c} = \max_t V_{b,c}^{(L-1)}[t]$. For the SHD dataset we used $a_{b,c} = \sum_t V_{b,c}^{(L-1)}[t]$

We also use two regularisation terms to constrain the spiking activity. These terms have been shown to not degrade the network performance [38]. First, we use a lower activity penalty to promote activity in the hidden layers.

$$\mathcal{L}_b^{low} = -\frac{\lambda^{low}}{N^{(l)}} \sum_{i}^{N^{(l)}} \left( \max(0, \nu^{low} - \zeta_{b,i}^{(l)}) \right)^2$$

Here $\zeta_{b,i}^{(l)} = \sum_t S_{b,i}^l[t]$ is the spike count of neuron $i$ in layer $l$ and batch sample $b$.

Secondly, the upper activity loss is added to prevent neurons to spike too often.

$$\mathcal{L}_b^{up} = -\lambda^{up} \max \left( 0, \frac{1}{N^{(l)}} \sum_{i}^{N^{(l)}} \zeta_{b,i}^{(l)} - \nu^{up} \right)$$

Finally the overall loss is obtained

$$\mathcal{L} = \mathcal{L}^{cross} + \frac{1}{B} \sum_{b=1}^{B} \left( \mathcal{L}_b^{low} + \mathcal{L}_b^{up} \right)$$

### E.5 Surrogate gradient

We use the following function as the surrogate derivate of the firing function defined in (19).

$$\frac{df(x)}{dx} = g(x) = \frac{1}{(\beta|x - V_{th}| + 1)^2}$$

In the sparse case we use

$$\frac{df(x)}{dx} = \begin{cases} g(x), & \text{if } |x - V_{th}| < B_{th} \\ 0, & \text{otherwise} \end{cases} \tag{20}$$

## E.6 Parameters

Here we summarise all parameters used in each dataset.

| | F-MNIST | N-MNIST | SHD |
|---|---|---|---|
| Number of Input Neurons | 784 | 1156 | 700 |
| Number of Hidden | 200 | 200 | 200 |
| Number of classes | 10 | 10 | 20 |
| Number epochs | 100 | 100 | 200 |
| B | 256 | 256 | 256 |
| T | 100 | 300 | 500 |
| $\Delta t$ | 1ms | 1ms | 2ms |
| $\tau$ hidden | 10ms | 10ms | 10ms |
| $\tau$ readout | 10ms | 10ms | 20ms |
| $\tau_{\text{eff}}$ | 20ms | N/A | N/A |
| $\theta$ | 0.2 | N/A | N/A |
| $V_r$ | 0 | 0 | 0 |
| $V_{rest}$ | 0 | 0 | 0 |
| $V_{th}$ | 1 | 1 | 1 |
| $B_{th}$ | 0.2 | 0.2 | 0.2 |
| $\beta$ | 100 | 100 | 100 |
| Optimiser | Adam | Adam | Adam |
| Learning Rate | 0.0002 | 0.0002 | 0.001 |
| Betas | $(0.9, 0.999)$ | $(0.9, 0.999)$ | $(0.9, 0.999)$ |
| $\lambda^{(low)}$ | 100 | 100 | 100 |
| $\nu^{(low)}$ | 0.001 | 0.001 | 0.001 |
| $\lambda^{(up)}$ | 0.06 | 0.06 | 0.06 |
| $\nu^{(up)}$ | 1 | 1 | 10 |

Table 3: Network and training parameters

# F  Other results

## F.1  Average activity for varying number of hidden neurons

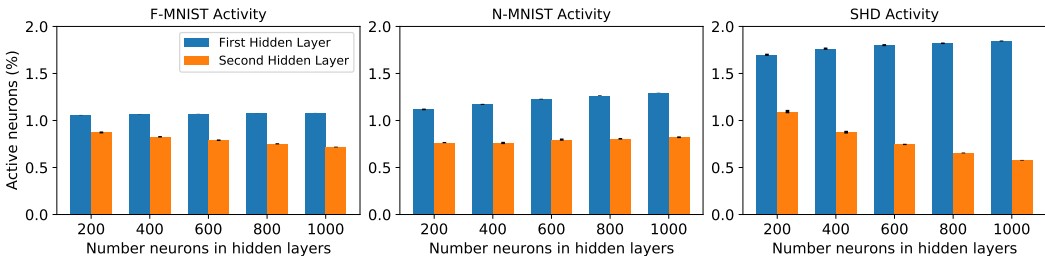

Figure 9: Average percentage of active neurons during training for each dataset and varying number of neurons (5 samples)

## F.2  Loss evolution for all datasets

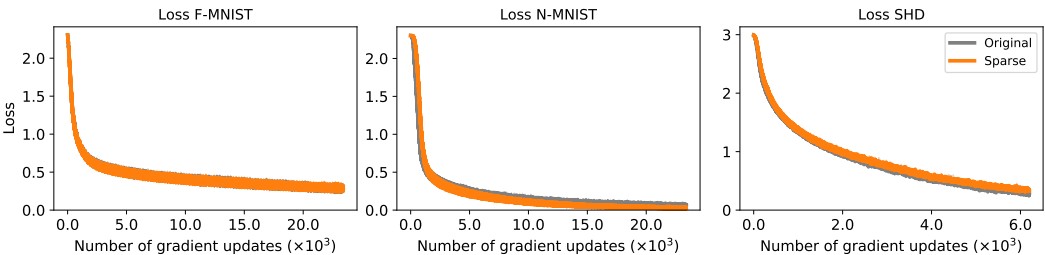

Figure 10: Loss for each dataset when using 200 hidden neurons (5 samples)

## F.3  Forward times

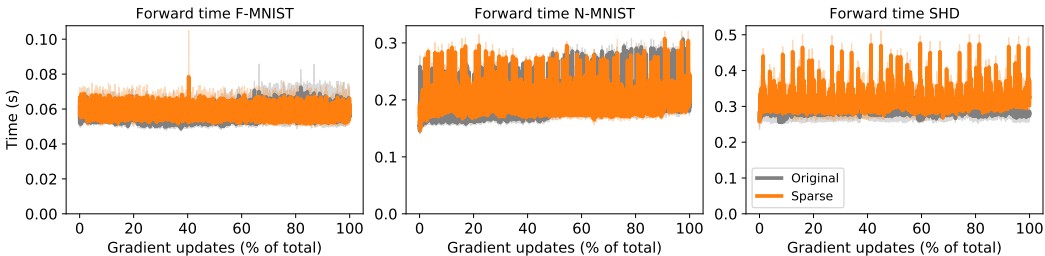

Figure 11: Forward time for each dataset when using 200 hidden neurons (5 samples)

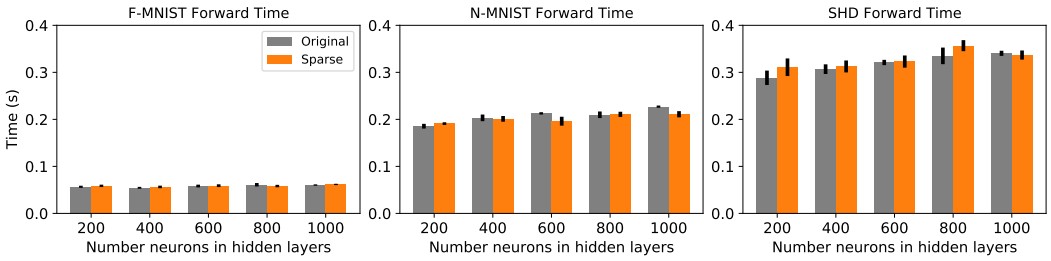

Figure 12: Average forward time for each dataset and varying number of neurons (5 samples)

## F.4  Backward times

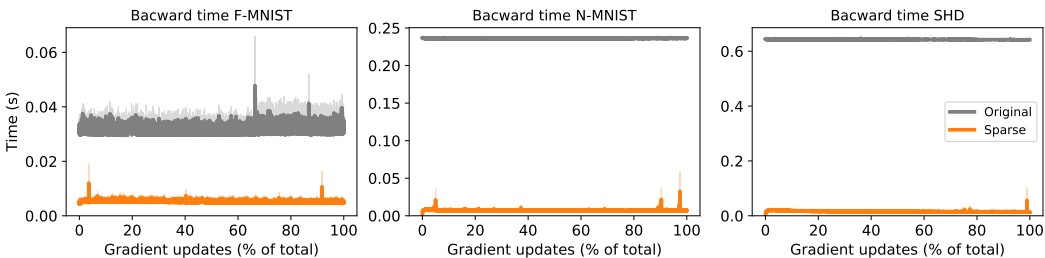

Figure 13: Backward time for each dataset when using 200 hidden neurons (5 samples)

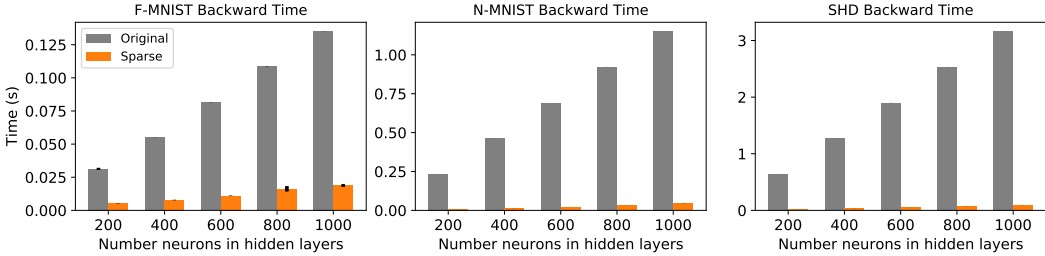

Figure 14: Average backward time for each dataset and varying number of neurons (5 samples)

## F.5 Forward memory

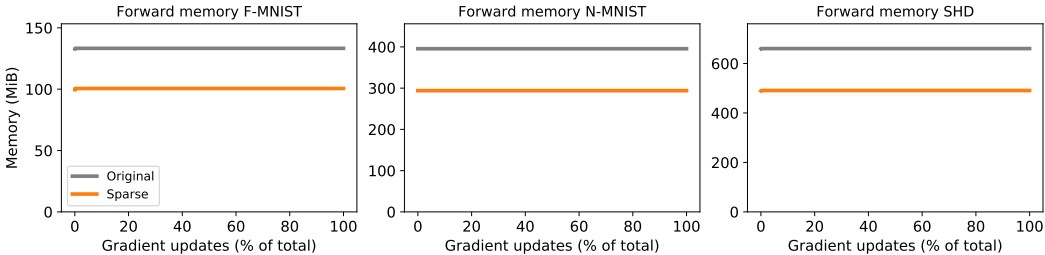

Figure 15: Forward memory for each dataset when using 200 hidden neurons (5 samples)

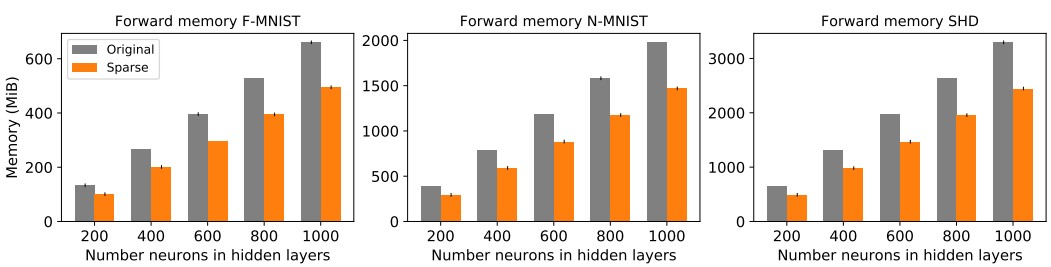

Figure 16: Average forward memory time for each dataset and varying number of neurons (5 samples)

## F.6 Backward memory

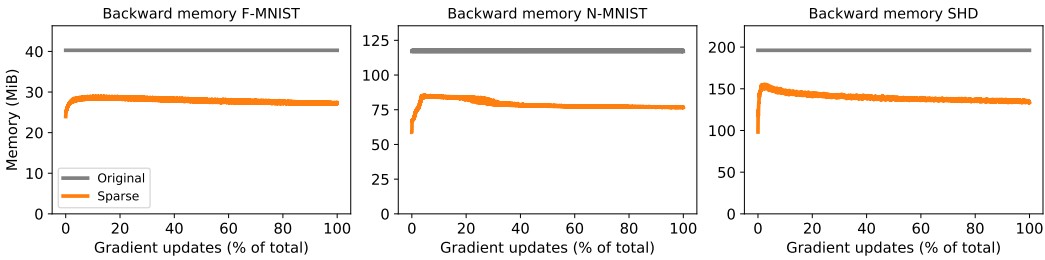

Figure 17: Backward memory for each dataset when using 200 hidden neurons (5 samples)

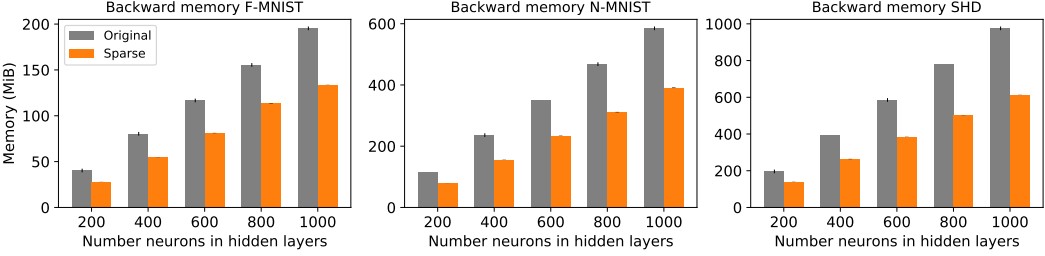

Figure 18: Average backward memory for each dataset and varying number of neurons (5 samples)

## F.7    Memory saved

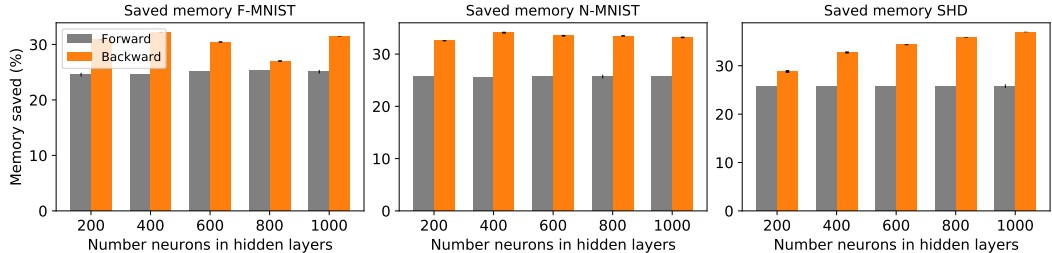

Figure 19: Average memory saved according to (14) for each dataset and varying number of neurons (5 samples)

## F.8    Results on different GPUs

Note that some entries are missing due to the GPU running out of memory.

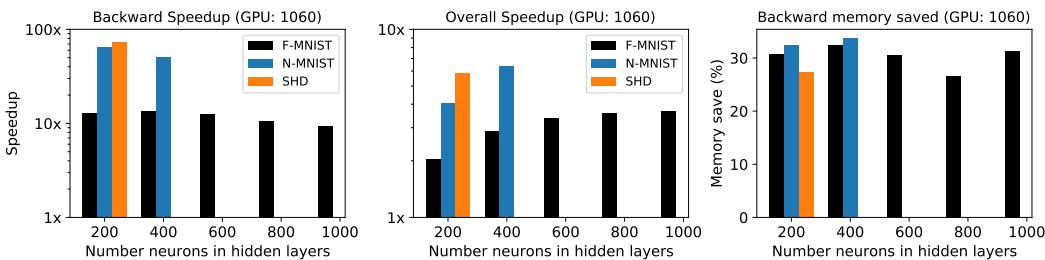

Figure 20: Backward speedup, overall layer speedup and backward memory memory saved on an NVIDIA GeForce GTX 1060

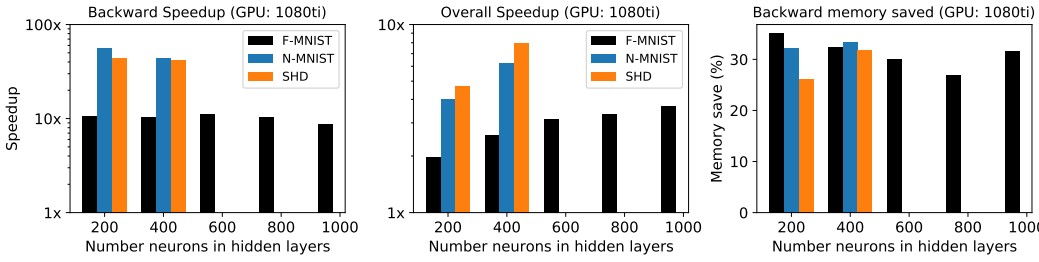

Figure 21: Backward speedup, overall layer speedup and backward memory memory saved on an NVIDIA GeForce GTX 1080 Ti

# G    Results on 5 hidden layers network

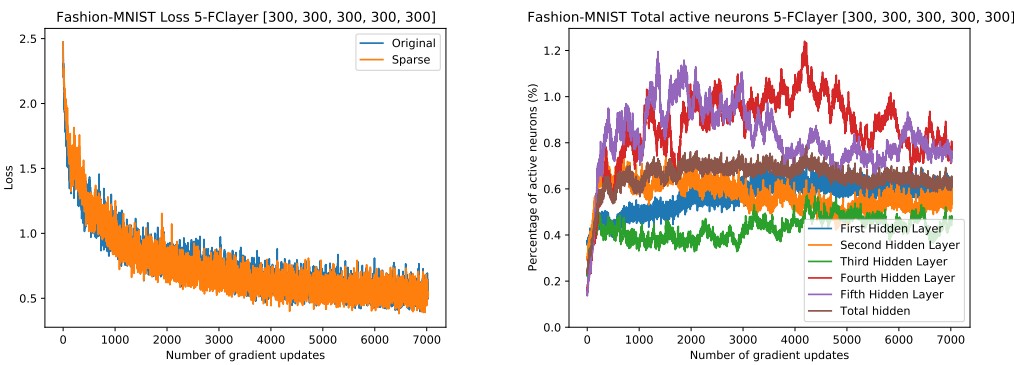

Figure 22: Loss history and layer activities when training on Fashion-MNIST with 5 hidden layers with 300 neurons each

# H    Weight gradient difference and relative error

We show here the difference between the original weight gradient and that obtained from our method as well as its relative error. We like to note that the original gradient is not necessarily more accurate than the sparse one as surrogate gradient training results in approximated gradients. Thus, it is not a ground truth gradient and the real test is done by showing that our method achieves the same or better loss and testing accuracy.

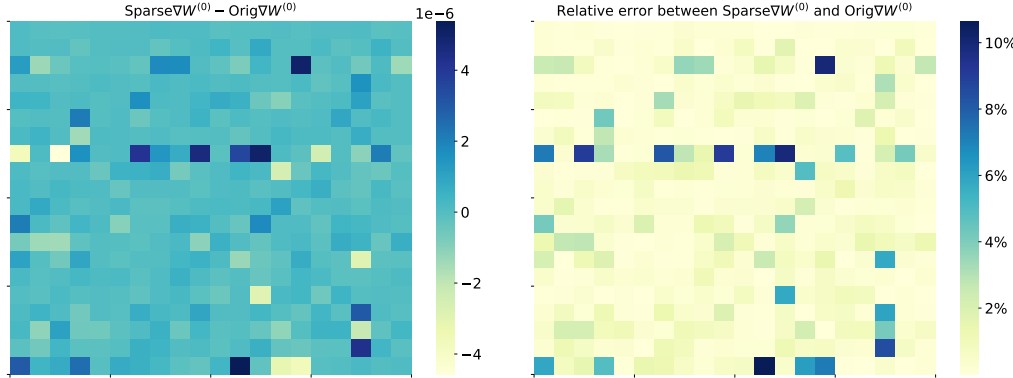

Figure 23: Difference between the weight gradients displayed in 3. The maximum relative error between two individual weights is $10.6\%$