# OpenReview forum: "Sparse Spiking Gradient Descent"
_NeurIPS.cc/2021/Conference — NeurIPS 2021 Poster_

### Official Review · Reviewer_DiNr · 2021-07-10

**Rating:** 6
**Confidence:** 4

**Summary:**

Surrogate gradient learning (SGL) is a recent approach to reconcile backpropagation with spiking neural networks. The approach works in discrete time. At each time step and for each neuron, a spike is generated if the potential is greater or equal than the threshold, and this is done by applying the Heaviside step function. This raises a problem when computing the gradients, as the derivative of Heaviside function is zero almost everywhere. The main idea of SGL is to replace this derivative by a "surrogate" one, typically the derivative of a sigmoid.

Here, the main idea of the authors is to compute the gradients only for the neurons whose potential is close to threshold, because other neurons have negligible gradients. Although very simple, this idea allows to reduce computation time by up to 70x, and memory by 40 on simple networks (MLP with 2 hidden layers), with no accuracy loss on simple datasets (Fashion-MNIST, Neuromophic-MNIST and Spiking Heidelberg Digits)

**Main Review:**

The idea is thus interesting, but it remains to be proven if it also works with deeper networks, convolutional layers, and with more complex datasets (eg ImageNet for static images or IBM DVS Gesture for neuromorphic datasets). For example I wonder whether the agreement between the approximated gradients and the true ones would still hold in deeper networks, or if errors would accumulate through the layers.

On the same ground, the accuracies that the authors obtained with the simple datasets are much below the SOTA (see for example Table 2 on Fang et al 2020 Incorporating Learnable Membrane Time Constant to Enhance Learning of Spiking Neural Networks). Of course this is partly due to the fact that the authors did not use convolutional layers. But it raises the question of whether their approach would still cause no accuracy loss on a network that really solves the tasks.


**Time Spent Reviewing:**

4

---

> ### Author Response · Authors · 2021-08-09
> **Response**
>
> ## 1. The idea is thus interesting, but it remains to be proven if it also works with deeper networks, convolutional layers, and with more complex datasets (eg ImageNet for static images or IBM DVS Gesture for neuromorphic datasets). For example I wonder whether the agreement between the approximated gradients and the true ones would still hold in deeper networks, or if errors would accumulate through the layers.
>
> We agree that there is no reason to believe that deeper architectures or other types of layers should not perform similarly. To further support this point we train a 6-layer fully connected network (5 hidden plus readout all with 300 neurons) on Fashion-MNIST. We measured the performance of the 5 hidden layers and we found a 15x faster backward time on the 1080-Ti GPU (better than the 10x speedup under these conditions with only 2 hidden layers as shown in Appendix F9) and a 25% memory improvement with a final accuracy of 82.7% after 30 epochs (slightly better than with 2 layers at 81.5%). The results are consistent with our previous findings and show that our approximation is robust to deeper architectures and the error is not carried out.
>
> Plots for these results can be found here: https://bit.ly/3ivRbNZ
>
> We also run a experiment including a convolutional and a pooling layer:
>
> [conv2D(filters=64, kernel=(3, 3), stride=1), LIF, pool((2, 2)), Linear(12544, 10), readout]
>
> on the Fashion-MNIST dataset. We obtained a test accuracy of 86.7% when using our approximation (clamping the gradients) and 86.9% without this approximation after 75 epochs.
>
> Plots for these results can be found here: https://bit.ly/3jKaqD0
>
> We do not report speedup improvements in this network as we have not developed a sparse convolutional kernel for this purpose and this falls out of the scope of our work. This results were obtained by running a dense implementation with sparse gradients (i.e. backpropagating dense tensors with over 99% entries corresponding to inactive neurons being zero). Thus, since a specific sparse convolution kernel was not used the speedup cannot be measured and this test only allows us to show that our gradient approximation also works for these layers.
>
> ## 2. On the same ground, the accuracies that the authors obtained with the simple datasets are much below the SOTA (see for example Table 2 on Fang et al 2020 Incorporating Learnable Membrane Time Constant to Enhance Learning of Spiking Neural Networks). Of course this is partly due to the fact that the authors did not use convolutional layers. But it raises the question of whether their approach would still cause no accuracy loss on a network that really solves the tasks.
>
> We believe that this is mainly due to the simplicity of the architecture used. On the particular case of the SHD dataset we have improved significantly by using a sum-over-time instead of a max-over-time loss as shown in [1]. We initially used the max-over-time loss to be consistent about the architecture used across datasets. We repeated the SHD experiments with the sum-over-time loss and we achieved an average of 74.3% accuracy which is the same as that achieved in [1] with the same network while obtaining identical results on speedup and memory improvement.
>
> We believe that the additional experiments we showed in the answer to Question 1 provide a compelling argument that indeed our gradient approximation holds for more complex networks.
>
>
> ## Summary of changes.
>
> We made the following changes to the paper:
>
> - We added the new experiments on the convolutional layers and deeper network.
>
> - We added the improved test accuracy on SHD using a different loss function
>
>
>
> [1] Zenke et al. The remarkable robustness of surrogate gradient learning for instilling complex function in spiking neural networks (2020)

---

### Official Review · Reviewer_B2Bg · 2021-07-12

**Rating:** 7
**Confidence:** 5

**Summary:**

This paper proposes a novel sparse gradient update rule to train spiking neural networks faster and more energy-efficient on GPUs. A mathematical derivation of backpropagation through time (BPTT) is given for a network of leaky integrate-and-fire (LIF), feed-forward, fully-connected layers without skip connections. A threshold is defined that marks neurons as active when they are close to the membrane threshold and the spike gradient is only propagated back if a neuron was active at that time during the forward pass. As a result, only a fraction of the actual update steps have to be computed, because all others would be zero anyway. The authors implement a version of their algorithm in CUDA to compare their sparse gradient update rule to the dense update rule from surrogate gradients. It is found, that on the three evaluated datasets, Fashion-MNIST, N-MNIST and Spiking Heidelberg Digits, a consistent improvement in memory efficiency and training speed can be measured while achieving similar accuracies.

**Limitations And Societal Impact:**

The limitations have not been discussed in detail. It is not clear why no deeper networks were used for the experiments. It is also not clear to me, why the speedup does not increase when increasing the layer size in Fig. 4.
Societal impact has not been discussed.

**Main Review:**

The results of this paper are an important contribution to the deep learning and neuromorphic community, as it tackles an important issue in the current state-of-the-art in the training of spiking neural networks. That is, the training speed is very slow, because potentially hundreds of time steps have to be simulated. As the common deep learning frameworks like PyTorch and TensorFlow offer no functionality to date, to compute sparse gradients efficiently, it is not possible to implement efficient update mechanisms for sparsely active networks like most spiking neural networks. The authors both, derived the theoretical foundation of sparse updates based on BPTT and the fact that the gradient is only non-zero close to the threshold and gave an actual implementation to validate their theory.

Still, the paper can be improved in a few aspects.
1.  Fig. 1 is not necessary in my opinion, as it does not provide any insights in the actual gradient updates, nor does it give an overview of the proposed algorithm. All relevant information can be found in Fig. 2.
2. In Tab. 2, the authors report theoretical savings in energy. As there is an actual implementation for GPUs, I think measuring the actual power consumption would be very interesting, to see if there is an actual gain in energy-efficiency.
3. In Fig. 3B, there is no colorbar and therefore it is not really possible to compare the results in a fair way. It could for example be, that all results are very close to zero and therefore the difference would be less impressive between the algorithms. Additionally, plotting the difference of Original GradW and Sparse GradW would make it easier to compare the two.
4. When reporting results in Fig. 3D, please also report some concrete results from the literature instead of just stating that "these accuracies are on par" (line 209).

Generally, it would have been interesting to see more results, especially on deeper neural networks. I see no imminent reason why the algorithm should not scale to deeper networks, on the opposite it should be even more efficient than surrogate gradients, that have already proven to be working with at least 10+ layers.

Overall, this paper shows solid results of a novel sparse gradient update rule with a small spiking neural network. I think this paper should be accepted, if at least some of the concerns in this review are addressed.


Small things that did not influence my score:
- ln 22: I would remove the "simultaneously", as it sounds like humans can solve auditory and visual recognition problems while also playing games, all at the same time
- Eq. 14 could be typeset prettier (without underscores and suitable variable names)
- Fig. 3D could be zoomed in more (e.g., start the bars at 25 %)
- A relevant paper to cite in the final version (was not published at submission): Event-based backpropagation can compute exact gradients for spiking neural networks, T. Wunderlich, C. Pehle, Sci Rep 11, 12829 (2021).

**Time Spent Reviewing:**

4.5

---

> ### Author Response · Authors · 2021-08-09
> **Response**
>
> ## 1. Fig. 1 is not necessary in my opinion, as it does not provide any insights in the actual gradient updates, nor does it give an overview of the proposed algorithm. All relevant information can be found in Fig. 2.
>
> The aim of this figure was to improve the readability of the overall paper. We are happy to move it to the supplementary or remove altogether if the reviewers believe the paper would be improved that way.
>
> ## 2. In Tab. 2, the authors report theoretical savings in energy. As there is an actual implementation for GPUs, I think measuring the actual power consumption would be very interesting, to see if there is an actual gain in energy-efficiency.
>
> We agree that measuring the actual energy consumption of the GPU would be ideal as opposed to just showing the theoretical upper limits. However, this is not straightforward to do, especially as we do not have physical access to the GPUs we used for this study (they are part of an external HPC cluster). Our results show that our method speeds up execution between one and two orders of magnitude, meaning that to a first approximation we would expect it reduce energy usage by the same factor. For example, for a 10x speedup we'd expect 10x less energy to be used. In practice, this is complicated by the fact that GPUs continue to use 30-50% of their peak power even when idle (see figure taken from [1]: https://bit.ly/3yzkPqZ). However, even in the worst possible case scenario where the original dense algorithm used the idle GPU power the whole time, and ours used the peak GPU power the whole time, with a 10x speedup total power saving would still be 3-5x for a 10x speedup given that the total computation time would be reduced.
>
>
> ## 3. In Fig. 3B, there is no colorbar and therefore it is not really possible to compare the results in a fair way. It could for example be, that all results are very close to zero and therefore the difference would be less impressive between the algorithms. Additionally, plotting the difference of Original GradW and Sparse GradW would make it easier to compare the two.
>
> We agree that we missed to add the fact that the weight gradients and spike gradients are on the same scale respectively. We added the colorbars to the figure. We also show here the difference between the weight gradients which is one order of magnitude smaller than the gradient values. We also computed that the maximum relative error between two individual weights is 9.6% as expected from the plots (plots can be seen in https://bit.ly/3CxT7gH).
>
> Our aim when plotting the gradients was to show that the direction of the gradient remains unchanged (the weights increase and decrease similarly in both cases). This is confirmed by the fact that we obtained the same loss throughout training as well as the same testing accuracy.
>
> ## 4. When reporting results in Fig. 3D, please also report some concrete results from the literature instead of just stating that "these accuracies are on par" (line 209).
>
> We will add the exact values in the final version of the paper
>
>
> ## 5. It would have been interesting to see more results, especially on deeper neural networks. I see no imminent reason why the algorithm should not scale to deeper networks.
>
> We agree that there is no reason to believe that deeper architectures or other types of layers should not perform similarly. To further support this point we train a 6-layer fully connected network (5 hidden plus readout all with 300 neurons) on Fashion-MNIST. We measured the performance of the 5 hidden layers and we found a 15x faster backward time on the 1080-Ti GPU (better than the 10x speedup under these conditions with only 2 hidden layers as shown in Appendix F9) and a 25% memory improvement with a final accuracy of 82.7% after 30 epochs (slightly better than with 2 layers at 81.5%). The results are consistent with our previous findings and show that our approximation is robust to deeper architectures and the error is not carried out.
>
> Plots for these results can be found here: https://bit.ly/3ivRbNZ
>
> ## Smaller comments
>
>  ### a. ln 22: I would remove the "simultaneously", as it sounds like humans can solve auditory and visual recognition problems while also playing games, all at the same time
>
> We will change the wording
>
>  ### b. Eq. 14 could be typeset prettier (without underscores and suitable variable names)
>
> We will change update it
>
>  ### c. Fig. 3D could be zoomed in more (e.g., start the bars at 25 %)
>
> We will add the exact values to avoid ambiguity
>
>  ### d. A relevant paper to cite in the final version (was not published at submission): Event-based backpropagation can compute exact gradients for spiking neural networks, T. Wunderlich, C. Pehle, Sci Rep 11, 12829 (2021).
>
> We will add it as part of the introduction on methods to train SNNs
>
>
>
> # Summary of changes.
>
> We made the following changes to the paper:
>
> - We will move/remove figure 1 if the reviewers believe it to be better that way
>
> - We added a comment on how a speedup directly translates into energy save due to GPUs consuming a lot of energy even when idle.
>
> - We added a colorbar to the plots and the difference of the weights gradients
>
> - We added the concrete accuracies obtained and baselines.
>
> - We added the experiment with deeper architecture
>
> - We updated the paper accordingly to the smaller comments
>
>
>
> [1] Knight et al. GPUs Outperform Current HPC and Neuromorphic Solutions in Terms of Speed and Energy When Simulating a Highly-Connected Cortical Model. Frontiers in Neuroscience 2018

---

### Official Review · Reviewer_X1Mr · 2021-07-13

**Rating:** 6
**Confidence:** 3

**Summary:**

This work introduces a new method to train spiking neural networks using far fewer gradients during the backwards back-propagation pass.  By only retaining the potential of so-called "active" neurons (those within a threshold epsilon of spiking) and learning through a surrogate function, these networks of spiking neurons can be sparsely trained (requiring only around 1% of the gradients).  The authors test these claims on three datasets (a time-encoded version of Fashion-MNIST, N-MNIST, and Spiking Heidelberg Digits), and achieve speedups of up to 70x and memory saving of 40% while maintaining approximately the same accuracy.  Further contributions interpret the learning rule from a Hebbian perspective and establish the savings in computational complexity.

**Ethical Concerns:**

This reviewer identifies no ethical concerns.



**Limitations And Societal Impact:**

This reviewer identifies no specific negative societal impacts from this work.



**Main Review:**

Originality: this work combines two disparate threads in neural network improvements: the literature on sparse learning (adaptive dropout and adaptive sparsity), and methods for surrogate learning for more effective spiking neural networks.  This is a combination I have not seen combined before, offering an unexpected and valuable insight to achieve efficient training.

Quality: the authors have provided a number of valuable contributions.  They provide a clear and detailed description of their neuron model, perform a computational complexity analysis, interpret and discuss their learning rule from a Hebbian perspective, and perform a number of high-quality experiments.  Multiple datasets are used (Fashion MNIST, Neuromorphic-NMIST, and the Spiking Heidelberg Dataset), and standard errors are evaluated.  Real-world computational efficiency and sparsity is evaluated, and GPU-based speedup and memory improvements are reported.

Limitations to this work were not sufficiently discussed here, however.  For example, one dataset used is Fashion-MNIST.  Typical performance scores are around ~90%+ for this dataset, while the methodology here achieves around 80%, though it is difficult to discern from the plot and no performance is stated.  This can be problematic if sparse spiking gradient descent is proposed as an alternative to standard training: the difference in performance must be small in order for this to be a significant work.  A variety of modified training rules or even only-approximately-correlated feedback [1] and surrogate learning rules can be effective, or successful on simpler datasets, but struggle to scale to reach state-of-the-art accuracy [2][3].

This is also true of more complex neuron architectures, whereas the architecture in this work is a three-layer fully-connected network.  As these are time-varying neurons, they are capable of quite complex computation, but an explicit evaluation against alternative architectures (LSTM, frame-based convnets, etc.) would allow a more realistic comparison point for the viability of sparse spiking gradients as a learning method.

Similarly, further baseline models or ablation studies would permit a better understanding of the sparse spiking gradient methodology.  For example, what if an equivalent random connected percentage (say 1-3%) of gradients were sampled and used from the full set of gradients?  Presumably, the most important gradients would be the ones that this work retains - those most relevant for around the spiking threshold - but this should be established in the work. How much worse would the method perform if a different set were chosen?  Additionally, as an ablation study, progressively widening the backpropagation threshold $Bth$ would provide valuable insight into the sparsity / accuracy tradeoff.

Finally, I would like to see a more detailed description of the time-stepped / recurrent implications of this model.  Fashion-MNIST is not a time-varying input set, so it is artificially coded into the time domain via spike latencies; N-MNIST has built-in movement of the MNIST digits; and the SHD codes represents its output in the time domain.  Presumably, a natively recurrent model like the one presented here would perform best compared to standard architectures that would not appropriately process the time-varying inputs, and allow training tasks which would be difficult with a standard static neuron model.  Depending on the time-binning, it could be cost-prohibitive to train a model in any way other than sparsely, and such an analysis could reinforce those advantages.

Clarity: The clarity in this manuscript is to be lauded.  The authors take care to draft meaningful, clear, and consistent figures.  Their derivations include color-coded term groupings to clearly discuss them in text that persist whenever term groups are discussed.  Sufficient details are reported for a reader to reproduce their work, and the real-world systems which they use are well described.

Significance: the primary motivation of this training method are to achieve improvements in efficiency and a reduction in computation, a topic of general interest.  Specifically, though, this method delivers substantial advantages for spiking applications, a more niche research area, with particularly strong applicability to the neuromorphic community.  In real-world speedups of training on spiking data, GPU training on the backward pass can be a substantial 70x faster, and be 40% more memory efficient.  These are valuable improvements, but these datasets are not yet representative of real-world usage or state-of-the-art computer vision, so the work is of more specific interest specifically to these niche communities.

Overall, this work contains multiple strong contributions, but the performance gap between state-of-the-art methods on the datasets and the performance measured here, the simpler neural network architectures, and the lack of a rigorous baseline to examine the impact of their simplification makes it challenging to assess whether future research can rely on the method introduced here to achieve sparse computation without sacrificing performance.

1.  Lillicrap, Timothy P., et al. "Random synaptic feedback weights support error backpropagation for deep learning." Nature communications 7.1 (2016): 1-10.
2. Lee, Jun Haeng, Tobi Delbruck, and Michael Pfeiffer. "Training deep spiking neural networks using backpropagation." Frontiers in neuroscience 10 (2016): 508.
3. Wu, Yujie, et al. "Spatio-temporal backpropagation for training high-performance spiking neural networks." Frontiers in neuroscience 12 (2018): 331.


----
After rebuttal:  I thank the authors for the interesting additional experimentation, and have raised my score in accordance with the more complex architectures and continued sustained performance.

**Time Spent Reviewing:**

4

---

> ### Author Response · Authors · 2021-08-09
> **Response (part 1/2)**
>
> ## 1.Limitations to this work were not sufficiently discussed here, however. For example, one dataset used is Fashion-MNIST. Typical performance scores are around ~90%+ for this dataset, while the methodology here achieves around 80%, though it is difficult to discern from the plot and no performance is stated. This can be problematic if sparse spiking gradient descent is proposed as an alternative to standard training: the difference in performance must be small in order for this to be a significant work. A variety of modified training rules or even only-approximately-correlated feedback and surrogate learning rules can be effective, or successful on simpler datasets, but struggle to scale to reach state-of-the-art accuracy.
>
> We would like to note, that we only showed the accuracies in our work to demonstrate that our approximation did not affect the final accuracy at all when compared to standard training on the same SNN (we refer to this standard setup as 'Original' in the labels). In other words, our aim was to show that our algorithm performs as well as current surrogate gradient descent methods on SNNs while doing less computations, executing faster and requiring less memory. Thus, the fact that we do not reach state-of-the-art accuracy in our experiments is due to using SNNs in the first place.
>
> We would also like to note that it is not the aim of our work to present spiking networks as a better alternative to standard training as that is a much broader subject that falls out of the scope of this work. Spiking neural networks have some advantages over standard neural networks such as their lower energy consumption when run on neuromorphic hardware which make them interesting as a potential alternative in certain applications. Our aim is to show that we can tackle to current computational bottleneck they face when training them on a GPU by using a sparse backpropagation algorithm.
>
>  A typical performance on the Fashion-MNIST dataset for standard training methods is above 90% accuracy. However, it is usually about 80-85% for spiking neural networks [1] and it can be further improved to 88% when training on spiking neural networks with trainable time constants [1] and to 94% when using deeper networks including convolutional, pooling, batch normalisation and dropout layers as well as learnable time constants [2]. We agree however, that the accuracy can be improved significantly on the SHD dataset by using a sum-over-time instead of a max-over-time loss as shown in [3]. We initially used the max-over-time loss to be consistent about the architecture used across datasets. We repeated the SHD experiments with the sum-over-time loss and we achieved an average of 74.3% accuracy which is the same as that achieved in [3] with the same network while obtaining identical results on speedup and memory improvement.
>
> ## 2. This is also true of more complex neuron architectures, whereas the architecture in this work is a three-layer fully-connected network. As these are time-varying neurons, they are capable of quite complex computation, but an explicit evaluation against alternative architectures (LSTM, frame-based convnets, etc.) would allow a more realistic comparison point for the viability of sparse spiking gradients as a learning method.
>
> Again, the aim of our work is not to show that spiking NNs are better than artificial NNs, but to show that for those who are already interested in SNNs, this is a method for making them run faster.
>
> With that said, we agree that further comparisons with more elaborate models would be ideal to put to test our work. We have done so with a convolutional architecture and a deeper fully connected network. Yet, there are other difficulties in doing so. First, benchmarking these models as simple as they seem is already a very compute intensive task. This is mainly due to the need to do a series of “warmup” runs to obtain accurate readings and to the fact that a two-layer+readout fully connected spiking network is actually a Tx3 fully connected network where T is the number of timesteps for the datasets. For instance, in the SHD case, the network has a total of 1500 layers. We estimate that just the results presented in the paper took a total of 3 months of GPU time to run.
>
> Secondly, given the lack of efficient sparse operators in most auto-differentiation platforms, every layer type requires to develop its own sparse CUDA kernel in order to be competitive with current heavily optimised libraries. In our work, we concentrated on implementing and testing this in spiking fully connected layers and we managed to show that sparse backpropagation is a lot faster and uses less memory. Developing these sparse kernels is not trivial and we are aware that having to do so for each layer is an important limitation of our work. Nevertheless, we would like to clarify that it is beyond the scope of our work to develop sparse operators for each layer type as we would be effectively reimplementing sparse versions of GPU libraries such as CuBLAS. Other libraries such as CuSPARSE, aimed at sparse operations, do not yet support tensor operations beyond 2 dimensions. This is no surprise since for now there is few reasons to believe that sparse tensor operations will significantly accelerate neural network performance. Our work aims to challenge this view and motivate the adoption of efficient sparse routines by showing for the first time that sparse BPTT can be faster while achieving the same accuracy as its dense counterpart.
>
> We add here a couple of experiments with more complex architectures, namely adding convolutional layers  and making the network deeper.
>
> Firstly, we trained a network consisting of the following layers:
>
> [conv2D(filters=64, kernel=(3, 3), stride=1), LIF, pool(2, 2), Linear(12544, 10), readout]
>
> on the Fashion-MNIST dataset. We obtained a test accuracy of 86.7% when using our approximation (clamping the gradients) and 86.9% without this approximation after 75 epochs.
>
> These results can be found here: https://bit.ly/3jKaqD0
>
> Secondly, we train a 6-layer fully connected network (5 hidden plus readout all with 300 neurons) on Fashion-MNIST. We measured the performance of the 5 hidden layers and we found a 15x faster backward time on the 1080-Ti GPU (better than the 10x speedup under these conditions with only 2 hidden layers as shown in Appendix F9) and a 25% memory improvement with a final accuracy of 82.7% after 30 epochs (slightly better than with 2 layers at 81.5%). The results are consistent with our previous findings and show that our approximation is robust to deeper architectures and the error does not increase with more layers making it suitable for more complex architectures.
>
> Plots for these results can be found here: https://bit.ly/3ivRbNZ
>
> Note: The additional results we presented using a convolutional layer were obtained by running a dense implementation with sparse gradients (i.e. backpropagating dense tensors with over 99% entries corresponding to inactive neurons being zero). Thus, since a specific sparse convolution kernel was not used the speedup cannot be measured and this test only allows us to show that our gradient approximation also works for these layers.
>
> ## 3. Similarly, further baseline models or ablation studies would permit a better understanding of the sparse spiking gradient methodology. For example, what if an equivalent random connected percentage (say 1-3%) of gradients were sampled and used from the full set of gradients? Presumably, the most important gradients would be the ones that this work retains - those most relevant for around the spiking threshold - but this should be established in the work. How much worse would the method perform if a different set were chosen? Additionally, as an ablation study, progressively widening the backpropagation threshold would provide valuable insight into the sparsity / accuracy tradeoff.
>
> We tested a random set of gradients instead of using those defined by the active neurons  on the SHD dataset and we found that learning does not take place at all (i.e. the testing accuracy is equal to the chance level, ~5%) as expected. We will add this results to the main text.
>
> We also trained the SHD dataset on the original 2-layer fully connected network with 400 neurons in each hidden layer with varying $B_{th}$. We found that the training is very robust to even extreme values of $B_{th}$. The loss and accuracy remains unchanged even when setting $B_{th}$=0.95 while converting what used to be a 40x speedup (using $B_{th}$=0.8) to  a nearly 200x speedup and from 55% to 85% memory improvement. We inspected the levels of activity while varying B_th and we found that there are enough active neuron to propagate the gradient effectively. As $B_{th}$ gets closer to 1 the number of active neurons decreases rapidly until there are no gradients to propagate.
>
>  We would like to thank the reviewer for their suggestion that lead to finding these very remarkable results.
>
> The plots for these results can be found here: https://bit.ly/3yM25EM

---

> ### Author Response · Authors · 2021-08-09
> **Response (part 2/2)**
>
> ## 4. Finally, I would like to see a more detailed description of the time-stepped / recurrent implications of this model. Fashion-MNIST is not a time-varying input set, so it is artificially coded into the time domain via spike latencies; N-MNIST has built-in movement of the MNIST digits; and the SHD codes represents its output in the time domain. Presumably, a natively recurrent model like the one presented here would perform best compared to standard architectures that would not appropriately process the time-varying inputs, and allow training tasks which would be difficult with a standard static neuron model. Depending on the time-binning, it could be cost-prohibitive to train a model in any way other than sparsely, and such an analysis could reinforce those advantages.
>
>  We agree with the reviewer that these are important points to consider. Our aim in choosing these datasets was on having a good representation of the main types of datasets that are usually used for testing SNNs. Namely, artificially encoding a dataset into spikes (Fashion-MNIST), reading spikes from a neuromorphic sensor such as a DVS camera (N-MNIST) and encoding complex temporally-varying data into spikes (SHD).
>
> One important thing to notice, as the reviewer points out, is that all these datasets are constrained in time to just a few hundred steps thus making the temporal dimension of their tensor representation no larger than 500 (at most up to 2000 in other work such as [1]). This is still small enough that it can be trained on dense implementations albeit at an important computational cost.
>
> The main advantage of our method is that it adapts to the level of activity of the network and it only computes gradients when they are non-negligible. For instance, it is common in spiking data to have no activity at the input (or small background noise) for important periods of time (see Appendix D which shows that this happens in all datasets). During these times there is no activity in the network and the gradients are zero; yet in a dense network implementation all this zeros are still being computed. With our algorithm, however, no gradient is computed at all during this times, as there is no activity in the network, thus saving computations and allowing for longer temporal dimension. In fact, with our method even if there is some small but meaningful activity in the network, our algorithm adapts to just computing the necessary gradients. For instance, in figure 7 in Appendix D, the activity decreases between 0.37 and 0.45 seconds thus resulting in lower activity in the network but yet meaningful information.
>
> This cost-prohibitive reality was evidenced when we attempted to run these experiments in smaller GPUs but some run out of memory for the dense models (see figures 20 and 21 in Appendix F.9). Thus,  sparse spiking backpropagation will allow to train data that runs for longer periods of time without requiring a more powerful hardware.
>
> We will add these considerations in the final version of the paper.
>
>
>
> ## Summary of changes.
> We made the following changes to the paper:
>  - We added the improved testing accuracies for our experiments by using more complex networks and/or different loss functions while keeping the performance advantages of our method.
>
>  - We clarified that the main limitation in our work lies in the lack of efficient sparse operators, thus requiring to resort to custom implementations.
>
>  - We added the results on varying $B_{th}$ and choosing a random subset of gradients.
>
>  - We elaborated more on the advantages of our method used on otherwise cost-prohibitive data.
>
>
>
> [1] Perez-Nieves et al. Neural heterogeneity promotes robust learning (2020)
>
> [2] Wei Fang et al. Incorporating Learnable Membrane Time Constant to Enhance Learning of
> Spiking Neural Networks (2020)
>
> [3] Zenke et al. The remarkable robustness of surrogate gradient learning for instilling complex function in spiking neural networks (2020)

---

> > ### Comment · Reviewer_X1Mr · 2021-08-28
> > **Thanks**
> >
> > I thank the authors for their detailed response to my questions.  With these additional experiments and results, I will raise my score to a 6.

---

### Official Review · Reviewer_8Wqn · 2021-07-16

**Rating:** 5
**Confidence:** 5

**Summary:**

The authors aimed at addressing the inefficient issue of SNN in backpropagation with dense tensor calculations. They designed a rule to estimate the spikes' gradient iteratively and theoretically avoid a lot of calculation.


**Main Review:**

Overall, this paper is written clearly and easy to follow. I have two major concerns about the current work.
1. First, the algorithm was only tested on small datasets. It is not clear how it would scale up to larger datasets like CIFAR10/100 and DVS-CIFAR.

2. As the author also stated, there lacked efficient sparse tensor operations in most auto-differentiation libraries. On the other hand, there will be an incompatibility between the sparse operation and the parallelism of GPU, which may intrinsically limit the potential application of the proposed methods to complex models. A related question is whether the proposed method can be supported by any existing neuromorphic hardware?

Another question to clarify is why the authors used the BP time of the second layer rather than the BP time of the whole network?

**Time Spent Reviewing:**

6

---

> ### Author Response · Authors · 2021-08-09
> **Response**
>
>  ## 1. First, the algorithm was only tested on small datasets. It is not clear how it would scale up to larger datasets like CIFAR10/100 and DVS-CIFAR.
>
>  These datasets seem small compared to those used for artificial NNs, but are close to the limit of what can be worked with for spiking NNs. The reason is that with surrogate gradient descent we are training using backprop through time (BPTT) and so we need to retain a copy in memory of the entire network state for each time step of the simulation. With this in mind, the SHD dataset runs for 500 time steps – much larger than the number of time steps used in a typical RNN – and therefore the effective dimensionality of the input data is 700 channels times 500 time steps = 350k. Our simulations are at the limit of what can fit in GPU RAM. This is a limitation of surrogate gradient descent, but not a limitation of our work which aims to substantially speed up this method. Proof of this is shown in Appendix F9 where smaller GPUs (1080Ti, 1060) run out of memory at 600 neurons when using the original dense algorithm.
>
> We would also like to point that the dimensionality of the SHD dataset is comparable in difficulty to datasets such as CIFAR10. Particularly, while the SHD dataset contains input spikes with dimensionality 700x500=350k and 20 different classes whereas CIFAR10 features have a dimensionality of 32x32x3=3072 and 10 classes. Even if we consider a conversion to spike times similar to the one we used for Fashion-MNIST we would end up with a dimensionality of 3072x100=302k.
>
> We would also like to note that when converting pixels values to spikes, the precise spike times do not carry as much information as in datasets where the precise spike timing is important for solving the task. This has been shown in [2] where a distinction is made between spatially constrained datasets (such as N-MNIST and spike converted Fashion-MNIST) and spatio-temporally constrained datasets (such as SHD). Thus, making the SHD task more challenging and interesting to solve in spiking networks. Our aim in choosing these datasets was on having a good representation of the main types of datasets that are commonly used for testing SNNs. Namely, artificially encoding a dataset into spikes (Fashion-MNIST), reading spikes from a neuromorphic sensor such as a DVS camera (N-MNIST) and encoding complex temporally-varying data such as speech into spikes (SHD).
>
> Further, the datasets we used are the ones typically used by the SNN community (for the reasons above), and therefore it makes sense to test our algorithms on what people in the target community are using.
>
> ## 2. As the author also stated, there lacked efficient sparse tensor operations in most auto-differentiation libraries. On the other hand, there will be an incompatibility between the sparse operation and the parallelism of GPU, which may intrinsically limit the potential application of the proposed methods to complex models. A related question is whether the proposed method can be supported by any existing neuromorphic hardware?
>
> There is indeed a lack of sparse tensor operators in most standard auto-differentiation libraries. Efficient ways to operate with sparse tensors are still an ongoing area of research that shows a lot of promise but has yet to be improved to be adopted in a general setting. Precisely for this reason we present our work as an advance towards this direction. Despite the inherent difficulties of implementing sparse operations on a GPU we managed to implement a sparse backpropagation algorithm on the GPU which is faster and more memory efficient by a considerable margin than a standard Pytorch implementation while at the same time without losing test accuracy.
>
> As for neuromorphic hardware, to the best of our knowledge, there is no neuromorphic hardware that supports a fully end-to-end backpropagation training. For this reason, most efforts towards using neuromorphic hardware consider off-chip training. There is some work however, that uses neuromorphic hardware for the forward pass and a GPU for the backward [1]. Thus rendering the backward pass the most time consuming and energy inefficient part of training. Consequently, this motivates the use of more efficient learning algorithms such as ours. Having said that, we believe that the idea of propagating a sparse gradient would be more suited for a potential neuromorphic backpropagation hardware since we effectively send the gradients in a sparse manner over time similarly to how spikes are sent in neuromorphic hardware during the forward pass.
>
>
> ## 3. Another question to clarify is why the authors used the BP time of the second layer rather than the BP time of the whole network?
>
> We report the improvements on the second layer because this is the only layer where we need to compute both gradients (weight and spike) in our network. In general, a neural network will have L spiking layers plus a readout layer which may not be spiking. The first layer will just need to compute the weight gradient while the remaining L-1 layers will have to compute both gradients.
>
> In our case, L=2 and thus we only have a single spiking layer that needs to compute both gradients (the second layer). A performance improvement in this layer translates to an improvement in all possible L-1 layers since all of them perform the same operation. Moreover, since the level of activity of the first hidden layer was similar to that of the second in all experiments (see Figure 1A) and it only requires computing the weight gradient, its computational cost was always less than that of the second layer, thus having a limited impact towards the whole network computational bottleneck and even more so in a deeper architecture. Finally, the readout and loss layers operate on a dense-tensor setup and thus their computational cost remains constant; consequently, we did not include them.
>
> To better support these points, we have run an additional experiment with a 6-layer fully connected network (5 hidden plus readout, each with 300 neurons) on Fashion-MNIST. We measured the performance of the 5 hidden layers and we found a 15x faster backward time on the 1080-Ti GPU (better than the 10x speedup under these conditions with only 2 hidden layers shown in Appendix F9) and a 25% memory improvement with a final accuracy of 82.7% after 30 epochs (slightly better than with 2 layers at 81.5%). The results are consistent with our previous findings and show that our approximation is robust to deeper architectures and the error in the gradient remains negligible even with several layers.
>
> Plots for these results can be found here: https://bit.ly/3ivRbNZ
>
>
> ## Summary of changes.
>
> We made the following changes to the paper:
> - We added a better overview of the relative complexity of the datasets used and how they compare to other datasets
>
> - We clarified that the main limitation in our work lies in the lack of efficient sparse operators, thus requiring to resort to custom implementations.
>
> - We clarified the role of using the second layer to measure performance and we will add the results on the deeper network experiment.
>
>
> [1] Cramer et al. Surrogate gradients for analog neuromorphic computing (2020)
>
> [2] Perez-Nieves et al. Neural heterogeneity promotes robust learning (2020)

---

### Decision · Program_Chairs · 2021-09-27

**Decision:**

Accept (Poster)

**Comment:**

Dear authors,

congratulations on your submission being accepted at Neurips.  The reviewers appreciated the sparse BP algorithm for speeding up training of spiking neural networks, and agree that it would be of interest to the Neurips audience. However, they also included feedback and constructive criticisms which we would ask you to include in your final version, in particular regarding limitations of the approach on large data sets due to memory limitations (linear scaling with number of steps). It would be ideal if you are able to include stronger empirical results on more complex data-sets in the final version.

Best, your AC